**SPECIAL ISSUE**
**CILIA AND FLAGELLA: FROM BASIC BIOLOGY TO DISEASE**

# Phosphatidylinositol 4,5-bisphosphate impacts extracellular vesicle shedding from *C. elegans* ciliated sensory neurons

**Malek W. Elsayyid\*, Alexis E. Semmel\*, Krisha D. Parekh, Nahin Siara Prova, Tao Ke and Jessica E. Tanis‡**

## ABSTRACT

Small secreted extracellular vesicles (EVs) mediate intercellular transport of bioactive macromolecules. How the membrane lipid phosphatidylinositol 4,5-bisphosphate [PI(4,5)P$_2$], which plays a crucial role in many cellular processes, impacts EV biogenesis is unclear. The primary cilium, a sensory organelle protruding from most non-dividing cells, transmits signals by shedding EVs called ectosomes. Here, we altered ciliary PI(4,5)P$_2$ in *C. elegans* by manipulating the expression of the type I phosphatidylinositol 4-phosphate 5-kinase (PIP5K1) PPK-1 and deletion of the phosphoinositide 5-phosphatase (INPP5E) *inpp-1*, then determined the impact on release of EVs that carried cargoes tagged with fluorescent proteins. We discovered that increasing PI(4,5)P$_2$ differentially affected ectosome shedding from distinct compartments, decreasing biogenesis of an EV subpopulation from the ciliary base, but enhancing budding from the cilium distal tip. Altering PI(4,5)P$_2$ levels also impacted the abundance and distribution of EV cargoes in the cilium, but not the sorting of the protein cargoes into distinct subsets of ectosomes. Finally, manipulating PI(4,5)P$_2$ did not affect cilium length, suggesting that changing PI(4,5)P$_2$ levels can serve as a mechanism to regulate ectosome biogenesis in response to physiological stimuli without impacting cilium morphology.

KEY WORDS: Cilia, Extracellular vesicle, Ectosome, *C. elegans*, PI(4,5)P2, PIP5K1, INPP5E

## INTRODUCTION

Non-motile primary cilia, sensory organelles enriched with receptors, ion channels and effector molecules, provide a platform to detect and transduce external signals that are important for development and cell homeostasis (Mill et al., 2023). More than 30 human diseases, termed ciliopathies, result from mutations in genes that encode proteins important for cilium structure or function (Mill et al., 2023). In addition to receiving signals, cilia can also transmit signals through the shedding of bioactive extracellular vesicles (EVs) (Luxmi et al., 2019; Wang et al., 2014; Wood et al., 2013). Cilia-derived EVs are utilized for multiple different purposes, including intercellular communication, transfer of signals between animals, regulation of ciliary signaling, and cell waste disposal (Cao et al., 2015; Luxmi and

Department of Biological Sciences, University of Delaware, Newark, DE 19716, USA.
*These authors contributed equally to this work

‡Author for correspondence ( jtanis@udel.edu)

 J.E.T., 0000-0002-7993-1013

King, 2022; Luxmi et al., 2019; Nager et al., 2017; Ojeda Naharros and Nachury, 2022; Wang et al., 2014, 2020; Wood et al., 2013). Thus, it is important to understand how EV biogenesis from this specialized organelle is regulated.

Contiguous with the plasma membrane, the periciliary membrane compartment (PCMC) at the ciliary base is separated from the cilium proper by the transition zone (TZ), which acts as a gate to confine proteins and lipids into the different compartments (Jensen et al., 2015; Mill et al., 2023; Park and Leroux, 2022). Distinct phosphoinositides, generated by reversible phosphorylation and dephosphorylation of the inositol head groups, bind specific proteins to spatially regulate ciliary processes and signaling (Balla, 2013; Conduit and Vanhaesebroeck, 2020). Phosphatidylinositol (4,5)-bisphosphate [PI(4,5)P$_2$] is concentrated in the plasma membrane and ciliary base, whereas phosphatidylinositol 4-phosphate (PI4P) is found at high levels in the cilium proper (Conduit and Vanhaesebroeck, 2020). This phosphoinositide compartmentalization is established by the action of type I phosphatidylinositol 4-phosphate 5-kinases (PIP5K1s), which generate PI(4,5)P$_2$ from PI(4)P (Conduit and Vanhaesebroeck, 2020; Loijens and Anderson, 1996; Weinkove et al., 2008), and the phosphoinositide 5-phosphatase INPP5E, which is enriched in the cilium proper and dephosphorylates PI(4,5)P$_2$ to produce PI(4)P (Chávez et al., 2015; Garcia-Gonzalo et al., 2015; Ukhanov et al., 2022). Loss of INPP5E function is the underlying cause of two ciliopathies, Joubert syndrome and MORM syndrome, as well as ciliopathy phenotypes in mice and zebrafish (Bielas et al., 2009; Jacoby et al., 2009), demonstrating the significance of phosphoinositide compartmentalization in cilia biology.

Stimulation of quiescent cultured cells with serum results in removal of INPP5E from the primary cilium, leading to accumulation of PI(4,5)P$_2$ in the cilium proper. The increase in PI(4,5)P$_2$ causes actin polymerization and excision of the cilium distal tip, a process known as cilia decapitation, which results in loss of the primary cilium and cell cycle re-entry (Phua et al., 2017). This excision is distinct from the ectocytosis of small EVs, termed ectosomes, that bud directly from the periciliary membrane compartment (PCMC) and cilium distal tip (Clupper et al., 2022; Nager et al., 2017; Ojeda Naharros and Nachury, 2022; van Niel et al., 2018; Wang et al., 2014, 2021). Whether biogenesis of ciliary ectosomes is regulated by PI(4,5)P$_2$ as observed for ciliary decapitation remains unknown.

Shedding of bioactive EVs from sensory neuron cilia can be observed in living *C. elegans* (Clupper et al., 2022; Lobo et al., 2025 preprint; Maguire et al., 2015; Razzauti and Laurent, 2021; Wang et al., 2014, 2015, 2021, 2024a). Ectosomes that bud from the cilium distal tip play a role in animal–animal communication and are deposited on the vulva during mating (Wang et al., 2014, 2020), whereas those shed from the PCMC of the ciliary base are taken up by surrounding glia or released into the environment (Clupper et al., 2022; Razzauti and Laurent, 2021). These two EV subpopulations, each with distinct signaling potentials, are differentially shed into

the environment in response to the physiological stimulus of mating partners (Clupper et al., 2022; Wang et al., 2021). Ectocytosis of EVs derived from the cilium tip, but not the ciliary base, relies on redundant kinesin-2 motors required for intraflagellar transport (IFT) (Clupper et al., 2022). Together, this indicates that distinct regulatory mechanisms govern EV shedding from the cilium proper versus the ciliary base in a spatially dependent manner.

Given the known impacts of $PI(4,5)P_2$ on protein localization, actin dynamics, endocytosis, exocytosis and membrane deformation (Katan and Cockcroft, 2020), we reasoned that the ciliary compartmentalization of $PI(4,5)P_2$ could have functional relevance for ectosome biogenesis. We found that in male-specific ray type B (RnB) EV-releasing neurons (EVNs), the *C. elegans* type I phosphatidylinositol 4-phosphate 5-kinase PPK-1 localizes to and regulates $PI(4,5)P_2$ abundance in the ciliary base, while the *C. elegans* phosphoinositide 5-phosphatase INPP-1 localizes to and impacts $PI(4,5)P_2$ in the cilium proper. Using a genetic approach to manipulate $PI(4,5)P_2$ abundance in ciliary compartments of worms expressing fluorescently labeled EV cargoes, we discovered that high $PI(4,5)P_2$ increases budding of EVs derived from the cilium tip but inhibits shedding of an EV subpopulation that comes from the PCMC. This altered EV biogenesis is not accompanied by a

change in cilium length, suggesting that increasing $PI(4,5)P_2$ in the cilium proper can serve as a mechanism to enhance EV ectocytosis from the distal tip without inducing ciliary decapitation.

## RESULTS

### Ciliary $PI(4,5)P_2$ is regulated by PPK-1 and INPP-1 in EV-releasing neurons

Like in other organisms, $PI(4,5)P_2$ has been shown to be depleted in the cilium proper of some *C. elegans* ciliated sensory neurons (DiTirro et al., 2019; Jensen et al., 2015), though the distribution of this phosphoinositide in the sensory cilia of the specialized EV-releasing neurons (EVNs) had not been explored. The cilia of the EVNs, which include the inner labial type 2 (IL2), male-specific ray type B (RnB), hook B (HOB) and cephalic male (CEM) neurons, have microtubules with specific structure and post-translational modifications as well as distinct motor proteins that enable abundant EV shedding from the cilium distal tip (Fig. 1A) (Maguire et al., 2015; O'Hagan et al., 2017; Silva et al., 2017). To determine whether these unique EVN cilia exhibit phosphoinositide compartmentalization, we used the *klp-6* promoter (Peden and Barr, 2005) to drive expression of a $PI(4,5)P_2$ reporter consisting of the pleckstrin homology (PH) domain of phospholipase C δ1 fused to

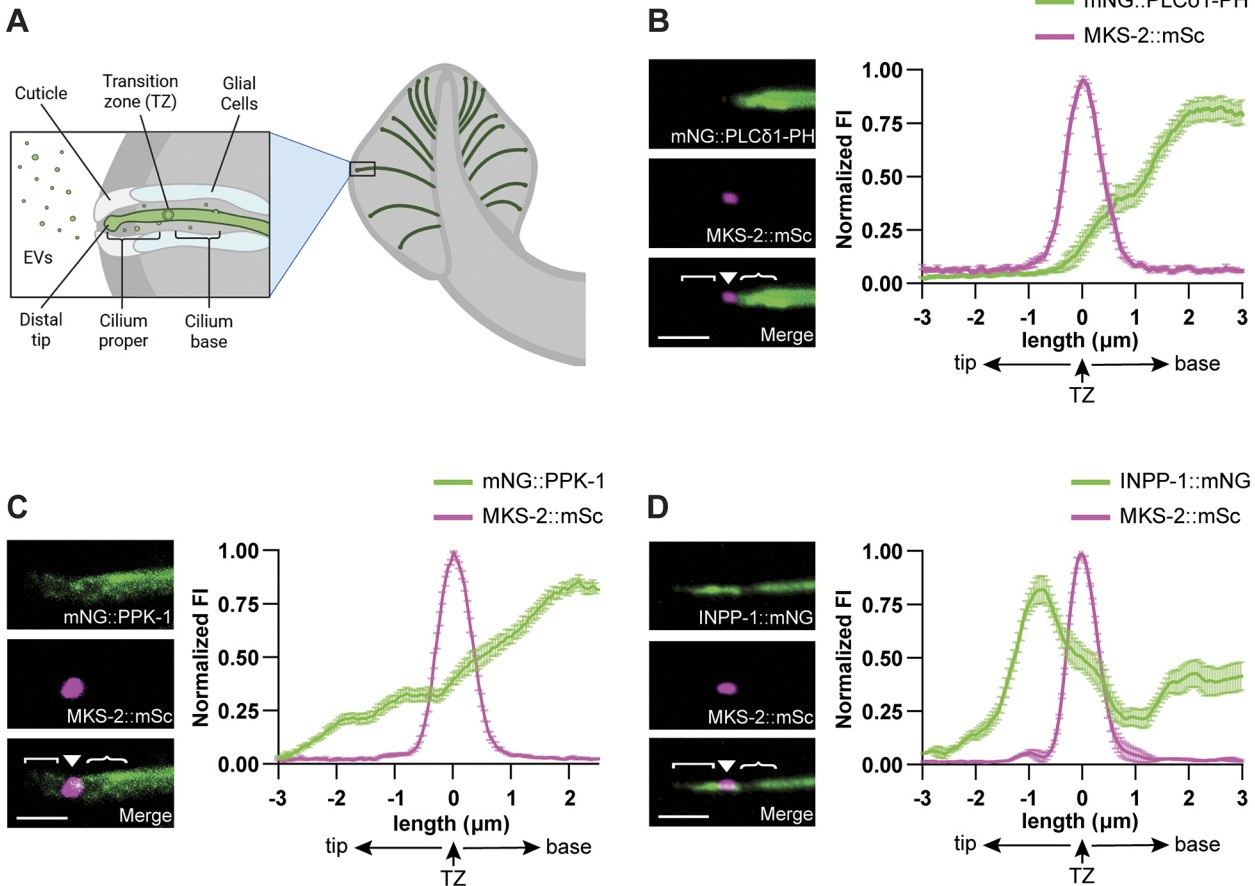

**Fig. 1. PPK-1 and INPP-1 are expressed in *C. elegans* male tail RnB sensory neurons.** (A) Schematic of EV shedding from the cilium distal tip and PCMC of the cilium base into the environment from a *C. elegans* RnB sensory neuron in the male tail. (B) In the control *him-5* mutant males, the $PI(4,5)P_2$ sensor mNG::PLCδ1-PH (top) is absent from the RnB cilium proper; MKS-2::mSc (middle) shows the transition zone (TZ). Right, normalized fluorescence intensity (FI) of mNG::PLCδ1-PH along *n*=28 RnB cilia. Average TZ length, 1.09±0.06 μm. (C) mNG::PPK-1 (top) and MKS-2::mSc (middle) in a RnB neuron. Normalized mNG::PPK-1 fluorescence intensity (right) shows abundant PPK-1 in the ciliary base with significantly less in the cilium proper; *n*=29 cilia. (D) INPP-1::mNG (top) localizes to the cilium proper, TZ (MKS-2::mSc; middle), and distal dendrite. INPP-1::mNG fluorescence (right; normalized) is reduced in the ciliary base; *n*=23 cilia. In all images (B–D), the cilium distal tip is oriented to the left; cilium proper ([), transition zone (▼) and ciliary base ({) are indicated. Scale bars: 2 μm. Data are represented as mean±s.e.m.

mNeonGreen (mNG::PLCδ1-PH) in the EVNs (DiTirro et al., 2019; Jensen et al., 2015). We co-expressed this integrated, multicopy mNG::PLCδ1-PH transgene with mScarlet-tagged MKS-2 (MKS-2::mSc), a protein in the Meckel syndrome (MKS) complex that localizes to the TZ (Lange et al., 2021). We found mNG::PLCδ1-PH localized to the distal dendrite and PCMC, which is similar to the ciliary pocket in mammals (Molla-Herman et al., 2010), but was excluded from the cilium proper of the RnB neurons (Fig. 1B). This demonstrates that the EVN cilia use the same phosphoinositide code as other sensory cilia.

In *C. elegans* the sole PIP5K1, PPK-1, generates $PI(4,5)P_2$ both *in vitro* and *in vivo* and is strongly expressed in the nervous system (Weinkove et al., 2008). GFP-tagged PPK-1 has been shown to have a localization pattern similar to that of GFP::PLCδ1-PH in the AWB sensory neurons, although this reporter was not expressed at endogenous levels or imaged with a TZ marker (DiTirro et al., 2019). To define where PPK-1 is localized in the EVNs, we inserted an N-terminal mNG tag on *ppk-1* (mNG::PPK-1) at the endogenous locus and isolated viable animals. This indicates that mNG::PPK-1 has at least a partial function, as complete loss of *ppk-1* results in lethality due to its essential role in embryogenesis (Weinkove et al., 2008). Co-expression of mNG::PPK-1 and MKS-2::mSc showed that PPK-1 was present in the dendrite and ciliary base, with significantly lower abundance in the cilium proper (Fig. 1C; Fig. S1A). We next examined the localization pattern of INPP-1, the *C. elegans* ortholog of INPP5E, which hydrolyses the 5-phosphate from $PI(4,5)P_2$. Our images of endogenous INPP-1 tagged with mNG at the C-terminus (INPP-1::mNG), co-expressed with MKS-2::mSc, showed abundant INPP-1 in the cilium proper and TZ, but a significant dip in abundance in the PCMC (Fig. 1D; Fig. S1B). Together, these data show that both PPK-1 and INPP-1 are expressed in the RnB neurons and that these $PI(4,5)P_2$ regulatory enzymes localize to distinct ciliary compartments in the EVNs.

We next sought to determine how altering levels of PPK-1 and INPP-1 impacted $PI(4,5)P_2$ abundance in the ciliary base and cilium proper. As *ppk-1* deletion mutants arrest as larvae, we instead used a transgene to overexpress PPK-1 in neurons (Weinkove et al., 2008). We found that this caused a significant increase in mNG::PLCδ1-PH fluorescence in the ciliary base, but not in the cilium proper (Fig. 2A–D). Given that $PI(4,5)P_2$ only increased where the mNG::PPK-1 endogenous reporter localized, this suggests that the effect is due to overexpression of the kinase. However, we cannot rule out the possibility that overexpression of PPK-1 also causes ectopic expression as it lacks a tag to study localization. Previously, loss of INPP5E in mammalian cells and loss of *inpp-1* in *C. elegans* AWB neurons has been shown to cause significant accumulation of $PI(4,5)P_2$ in primary cilia (DiTirro et al., 2019; Ukhanov et al., 2022). Consistent with this, we observed mNG::PLCδ1-PH in the cilium proper of *inpp-1* mutant RnB neurons, but not wild-type animals (Fig. 2A–C). This indicates that PPK-1 and INPP-1 both act to partition $PI(4,5)P_2$ in the EVNs (Fig. 2E).

### Altering $PI(4,5)P_2$ in the ciliary base impacts ectosome shedding from this compartment

Having established that overexpression of PPK-1 causes an increase in $PI(4,5)P_2$ in the ciliary base (Fig. 3A), we sought to determine the impact on EV shedding from this compartment. The $Ca^{2+}$ homeostasis modulator ion channel CLHM-1 (Tanis et al., 2013) and TRP polycystin ion channel PKD-2 (Barr and Sternberg, 1999) are EV cargoes that have been identified through *in vivo* imaging (Clupper et al., 2022; Wang et al., 2014) as well as in an EV proteomic dataset (Nikonorova et al., 2022). These proteins are found in discrete EV subpopulations, with CLHM-1 packaged into

EVs shed from the ciliary base and PKD-2 into EVs predominantly shed from the cilium distal tip (Clupper et al., 2022; Wang et al., 2021). To quantify the amount of EVs released into the environment from male tail EVNs, we use total internal reflection fluorescence (TIRF) microscopy to image animals expressing tdTomato-tagged CLHM-1 (CLHM-1::tdT) and GFP-tagged PKD-2 (PKD-2::GFP) single-copy transgenes (Clupper et al., 2022). The *him-5(e1490)* mutation, which causes increased frequency of X chromosome nondisjunction (Hodgkin et al., 1979), is in the genetic background of all strains analyzed to generate male offspring. Previously, we used lambda spectral imaging followed by linear unmixing to demonstrate that CLHM-1::GFP and PKD-2::GFP signal in EVs released into the environment is bona fide GFP emission and not background (Clupper et al., 2022). Here, we show that fluorescent EV-like signals are not detected in *him-5(e1490)* control animals imaged and analyzed under the same conditions as those expressing PKD-2::GFP and CLHM-1::tdT (Fig. S3A,B). These EVs are likely ectosomes, as individual EVs appear to bud directly from the plasma membrane (Fig. S3C,D); multivesicular bodies (MVBs) have not been observed in EVN cilia, and MVB components are not required for shedding (Clupper et al., 2022; Wang et al., 2014, 2024b).

We crossed the neuronal PPK-1 overexpression transgene with the CLHM-1::tdT and PKD-2::GFP single-copy transgenes and discovered that overexpression of PPK-1 caused a decrease in shedding of the ciliary base-derived CLHM-1-containing EVs, but had no impact on the release of distal tip-derived PKD-2 EVs (Fig. 3B–E; Fig. S4A,B). We next examined the colocalization of the ion channel cargoes in EVs and found that the probability of PKD-2::GFP being in a CLHM-1::tdT EV was unchanged by PPK-1 overexpression (Fig. 3F). This suggests that although overall shedding of the CLHM-1-containing ectosomes is reduced, the sorting of cargoes into these EVs is not disrupted. We then sought to determine how loss of PPK-1 function affects CLHM-1 EV biogenesis. Null mutations in *ppk-1* cause embryonic lethality (Weinkove et al., 2008) and no viable *ppk-1* mutants have been isolated. However, we discovered that the mNG::PPK-1 endogenous reporter strain exhibited significantly reduced brood size and embryonic viability, suggesting that the addition of mNG to the N-terminus of PPK-1 results in a partial loss of function (Fig. 3G,H). We crossed mNG::PPK-1 with the CLHM-1::tdT transgene and found that this reduced PPK-1 function led to a significant increase in the shedding of CLHM-1-containing EVs (Fig. 3I). This suggests that both increases and decreases in PPK-1 activity affect the release of base-derived CLHM-1 EVs into the environment.

### Overexpression of PPK-1 alters ciliary distribution and abundance of EV cargoes

$PI(4,5)P_2$ can impact the ciliary abundance of receptors due to its roles in endocytosis and trafficking (De Craene et al., 2017; DiTirro et al., 2019; Mukhopadhyay et al., 2010; Posor et al., 2015). To explore whether increased $PI(4,5)P_2$ in the ciliary base had an effect on localization or enrichment of the EV cargoes in the RnB neurons, we crossed the PPK-1 overexpression transgene with CLHM-1::tdT and MKS-2::mNG, then quantified CLHM-1::tdT fluorescence intensity and volume in the different ciliary compartments (Fig. S2B). Whereas CLHM-1 was present in the ciliary base of all control RnB neurons analyzed, 34% of the RnBs in the PPK-1 overexpressor lacked any detectable CLHM-1 in this compartment (Fig. 4A–C). Overall, PPK-1 overexpression significantly reduced CLHM-1::tdT abundance in the ciliary base, but had minimal impact on CLHM-1 in the cilium proper (Fig. 4B,C).

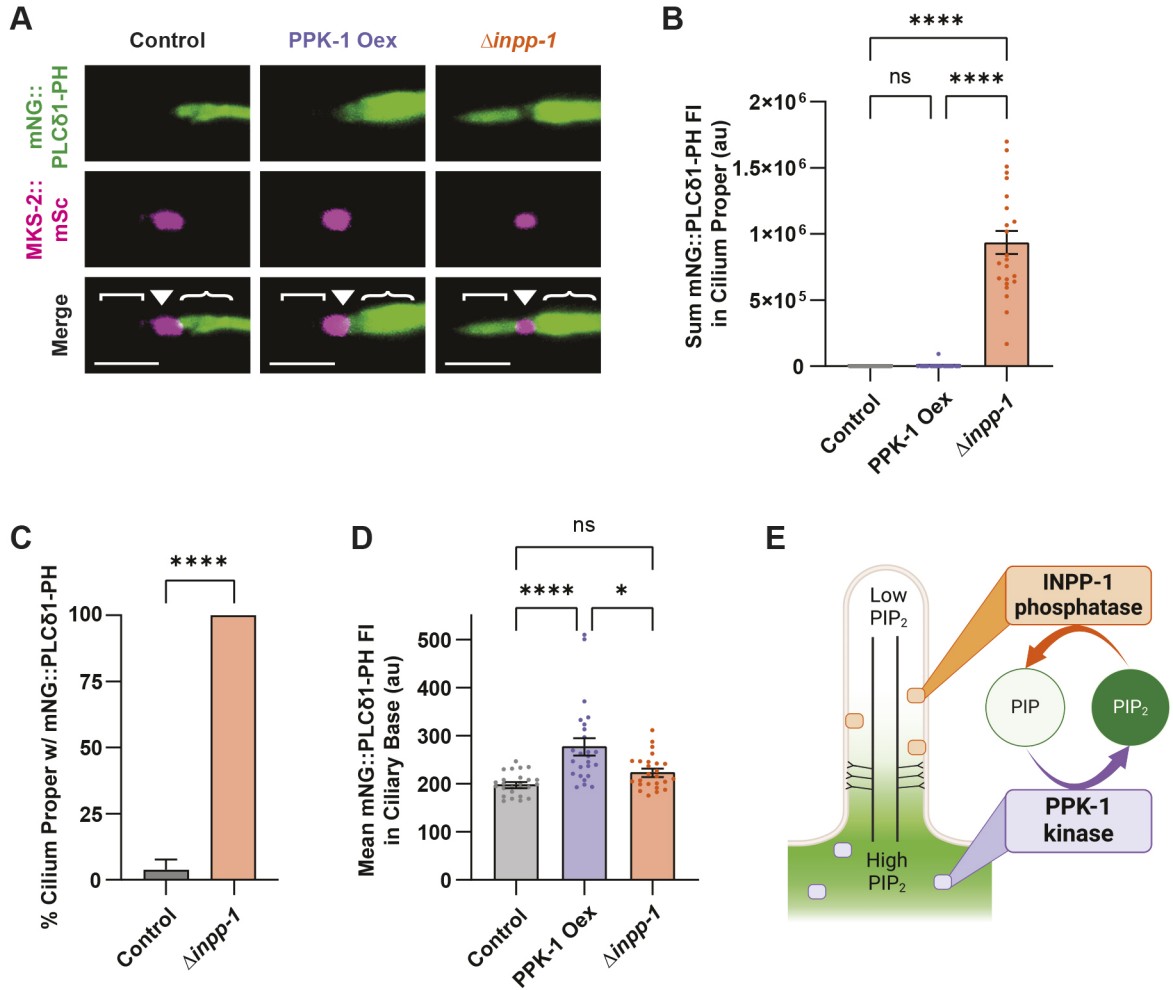

**Fig. 2. PPK-1 and INPP-1 regulate PI(4,5)P$_2$ abundance in RnB sensory neuron cilia.** (A) mNG::PLCδ1-PH (top) and MKS-2::mSc (middle) in wild-type, PPK-1 overexpression (Oex) and *inpp-1(gk3262)* loss of function mutant (Δ*inpp-1*) animals. Identical settings used for all images; cilium distal tip is oriented to the left, cilium proper ([), transition zone (▼) and ciliary base ({) are indicated. Scale bars: 2 μm. (B) Sum intensity of mNG::PLCδ1-PH in the cilium proper for control (gray), PPK-1 overexpressor (purple), and *inpp-1* mutant (orange) animals. *inpp-1* mutants showed a significant increase in mNG::PLCδ1-PH in the cilium proper compared to the control; $n \geq 26$ cilia. (C) Percentage of RnBs with mNG::PLCδ1-PH in the cilium proper; PI(4,5)P$_2$ was present in all 23 *inpp-1* mutant RnB cilia analyzed. (D) mNG::PLCδ1-PH fluorescence intensity in the ciliary base of control, PPK-1 overexpressor, and *inpp-1* mutant animals. Measurement of mean fluorescence in the ciliary base, defined as a ROI within 3 μm of the TZ, showed that PPK-1 overexpression increases the abundance of PI(4,5)P$_2$ in the ciliary base; $n = 25$. (E) Diagram showing INPP-1 and PPK-1 localization as well as how these enzymes affect ciliary PI(4,5)P$_2$ levels in wild type animals. Data are represented as mean±s.e.m. *$P < 0.05$, ****$P < 0.0001$; ns, not significant [Kruskal–Wallis test with Dunn's multiple comparisons was used (B,D); binomial test (C)]. au, arbitrary units.

Although low CLHM-1 in the ciliary base could explain the decrease in shedding of ectosomes with this cargo (Fig. 3E), we note that a reduction in CLHM-1 abundance in the PCMC does not necessarily correlate with a decrease in the number of shed EVs (Clupper et al., 2022).

Internalization of channels via clathrin-mediated endocytosis is dependent on PI(4,5)P$_2$, after which, receptors are sorted for recycling or endosomal degradation (De Craene et al., 2017; Posor et al., 2015). Therefore, we wanted to determine whether enhanced degradation could explain the reduced CLHM-1 EV shedding and ciliary base abundance in animals overexpressing PPK-1. The hepatocyte growth factor regulated tyrosine kinase–substrate signal transducing adaptor molecule (HRS–STAM) complex binds to and sorts ubiquitylated membrane proteins on the early endosome to the MVB for lysosomal degradation (Bache et al., 2003; Lloyd et al., 2002; Mizuno et al., 2003; Raiborg et al., 2002). The *C. elegans* STAM ortholog, STAM-1, is expressed in the EVNs and PKD-2::GFP accumulates in early endosomes in the ciliary base

and distal dendrite of *stam-1* mutant males, suggesting that STAM-1 sorts PKD-2 for lysosomal degradation (Hu et al., 2007). To determine whether STAM-1 also promotes the degradation of CLHM-1, we crossed the *stam-1(ok406)* null allele with CLHM-1:: tdT and analyzed ciliary abundance. Loss of *stam-1* did not alter the amount of CLHM-1 in the ciliary base, the overall distribution of CLHM-1 in the RnB neurons or CLHM-1 EV shedding (Fig. S5A–D). Interestingly, release of PKD-2::GFP EVs was also unaffected by loss of *stam-1*, suggesting that reduced lysosomal degradation of PKD-2 does not enhance shedding of EVs from the ciliary tip (Fig. S5E). Whereas the HRS–STAM complex acts on early endosomes to sort certain membrane proteins, including PKD-2, to the MVB for lysosomal degradation, our results suggest that a different mechanism is used to downregulate ciliary CLHM-1. Additional experimentation will be required to identify specific mutants that disrupt the endocytosis and degradation of CLHM-1.

Although PPK-1 overexpression did not impact PKD-2 ectosome biogenesis, analysis of ciliary PKD-2::GFP in these animals showed

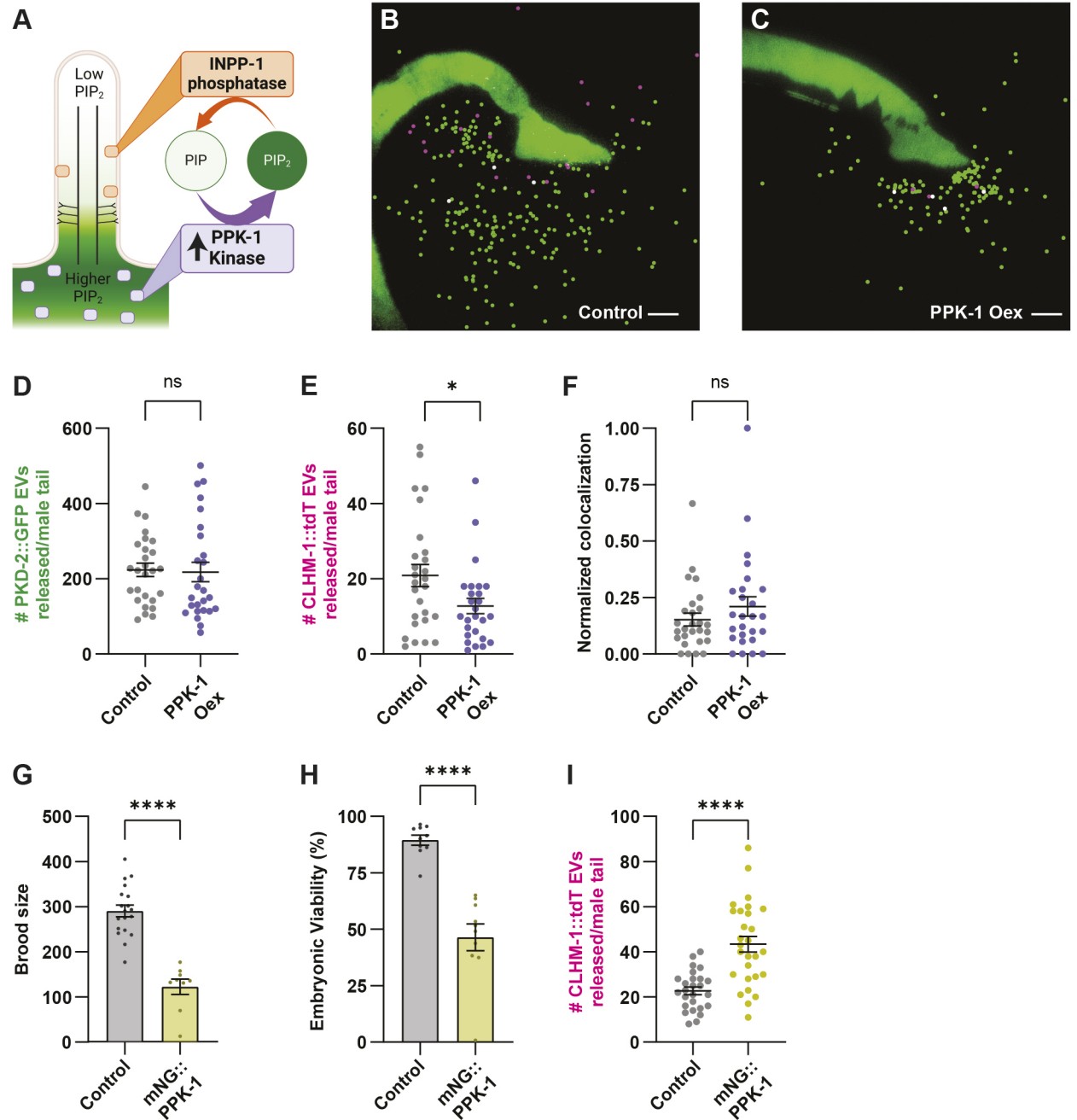

**Fig. 3. Both overexpression of and loss of PPK-1 impacts shedding of CLHM-1-containing EVs.** (A) Schematic depicting higher ciliary base PI(4,5)P$_2$ in PPK-1 overexpression (Oex) animals. (B,C) Representative images of CLHM-1::tdT (*henSi3*) and PKD-2::GFP (*henSi20*) EVs released from (B) control and (C) PPK-1 Oex animals. EVs are marked with spots (Imaris software) for visualization; original, non-cropped, unmarked images are in Fig. S4. Scale bars: 10 μm. (D) PKD-2::GFP EV release is not affected by PPK-1 Oex. (E) Release of CLHM-1::tdT EVs decreases in PPK-1 Oex animals. (F) Probability of PKD-2::GFP presence in a CLHM-1::tdT EV is unchanged by PPK-1 Oex. For D–F, $n \geq 26$. (G,H) Brood size (G) and embryonic viability (H) is significantly reduced in animals with the mNG::PPK-1 endogenous reporter (yellow) compared to the control (gray); $n \geq 9$. (I) Release of CLHM-1::tdT EVs increases in mNG::PPK-1 animals, which have partial loss of PPK-1 function; $n \geq 26$. Data are represented as mean±s.e.m. *$P < 0.05$; ****$P < 0.0001$; ns, not significant (two-tailed Mann–Whitney test).

that high PI(4,5)P$_2$ in the PCMC disrupted the relative abundance of PKD-2 in the different ciliary compartments (Fig. 4D–F). PI(4,5)P$_2$-binding Tubby (TUB) and Tubby-like (TULP) proteins link ciliary membrane proteins to the intraflagellar transport (IFT) system to mediate trafficking (DiTirro et al., 2019; Mukhopadhyay et al., 2010). Thus, the observed increase in PKD-2::GFP in the TZ and decrease in the PCMC suggests that PPK-1 overexpression could impact on trafficking of PKD-2 to the cilium.

## Reduced PPK-1 function does not impact CLHM-1 abundance in the RnB neurons

After evaluating the effect of PPK-1 overexpression on the abundance and distribution of EV cargoes in the RnBs, we considered that the increase in CLHM-1 EV shedding observed in the mNG::PPK-1 animals, which have reduced PPK-1 function, could also be associated with altered CLHM-1 ciliary abundance. To test this, we crossed mNG::PPK-1 with CLHM-1::tdT and MKS-2::mNG and

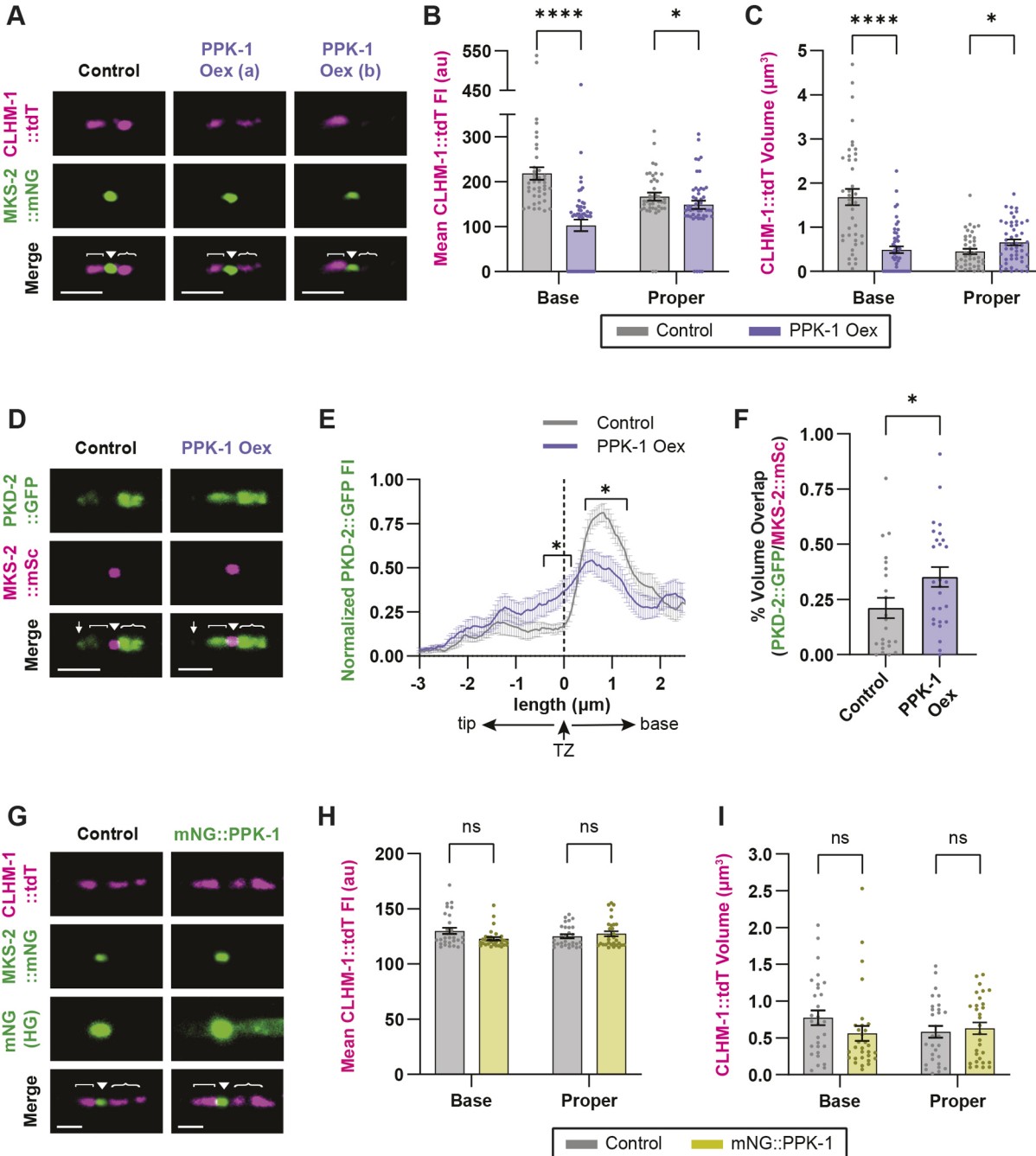

**Fig. 4. PPK-1 overexpression affects the abundance and localization of EV cargoes in the RnB neurons.** (A) CLHM-1::tdT (top) and MKS-2::mNG (middle) localization in cilia of control and PPK-1 overexpression (Oex) animals. Cilium proper oriented to the left, two representative images shown for the PPK-1 Oex (a,b) to show the different phenotypes observed. (B,C) CLHM-1::tdT mean fluorescence intensity (FI) (B) and volume (C) in the ciliary base and cilium proper in control (gray) and PPK-1 Oex (purple) animals. 17 out of 50 PPK-1 Oex cilia analyzed lacked CLHM-1::tdT in the ciliary base; CLHM-1::tdT was present in the base for all control cilia (n=41). (D) PKD-2::GFP (top) and MKS-2::mSc (middle) in control and PPK-1 Oex animals. (E) Quantification of PKD-2::GFP fluorescence intensity shows altered localization in the TZ and ciliary base of PPK-1 Oex animals compared to the control. Dotted line indicates center of the TZ, *P<0.05 for all points indicated, n≥14. (F) Percentage of MKS-2::mSc TZ volume that overlaps with PKD-2::GFP; n≥23. (G) CLHM-1::tdT (top) and MKS-2::mNG (middle); same images presented below with altered settings (high green; HG) to show the absence (control, left) versus presence of the mNG::PPK-1 endogenous reporter (right). (H,I) Mean fluorescence intensity (H) and volume (I) of CLHM-1::tdT in the ciliary base and cilium proper in control (gray) and mNG::PPK-1 animals (yellow), which have partial loss of PPK-1 function; n≥29. Note, CLHM-1::tdT control values in B,C differ from the values in H,I because imaging was conducted over 2 years apart by separate individuals on different Andor Dragonfly microscopes. In this figure, all images are labeled as cilium distal tip (↓), cilium proper ([), transition zone (▼) and ciliary base ({). Scale bars: 2 µm. Data are represented as mean±s.e.m. *P<0.05, ****P<0.0001; ns, not significant (two-tailed Mann–Whitney test). au, arbitrary units.

quantified CLHM-1::tdT fluorescence. Surprisingly, the intensity and volume of CLHM-1::tdT in the ciliary base and cilium proper was not significantly different between the mNG::PPK-1 and control animals

(Fig. 4G–I). Furthermore, the relative distribution of CLHM-1::tdT between the cilium proper and ciliary base was not affected by reduced PPK-1 function (Fig. 4G; data not shown). This suggests that

rather than simply regulating the abundance and localization of EV cargoes within the neurons from which EVs are shed, PI(4,5)P$_2$ can more directly inhibit ectocytosis from the ciliary base.

## Elevated PI(4,5)P$_2$ in the cilium proper increases EV shedding from the distal tip

Unlike INPP5E mutant mice, which die soon after birth, *inpp-1* mutant *C. elegans* appear essentially normal, despite accumulation of PI(4,5)P$_2$ in the cilium proper (Fig. 2A–C; Fig. 5A). We created *inpp-1* null mutants that express the CLHM-1::tdT and PKD-2::GFP single-copy transgenes to assess how high PI(4,5)P$_2$ in the cilium impacts EV release. Loss of *inpp-1* significantly increased the shedding of the tip-derived PKD-2-containing ectosomes from the RnBs (Fig. 5B–E; Fig. S4C,D). However, the average number of CLHM-1 ectosomes as well as the probability of PKD-2::GFP being present in a CLHM-1::tdT EV was unchanged between the control and *inpp-1* mutant (Fig. 5F,G). We note that the effect of INPP-1 on cilium tip versus base-derived EV shedding was recently confirmed by another group using the tetraspanin TSP-6 EV cargo (Lobo et al., 2025 preprint). This suggests that high PI(4,5)P$_2$ in the cilium proper enhances EV shedding specifically from the distal tip.

Given this impact of INPP-1 on PKD-2 EV shedding, we next examined PKD-2::GFP distribution in RnB neuron cilia at two different time points. Although there was no difference in PKD-2

ciliary localization in the *inpp-1* mutant right after mounting the animals for imaging, after 40 min, we found that PKD-2 was absent from the cilium proper and distal tip in 28% of the *inpp-1* mutant RnB cilia examined, a significantly higher percentage than observed for wild type (Fig. 6A,B). Analysis of only the cilia that initially contained PKD-2 showed that 20% in the *inpp-1* mutants lost PKD-2 signal entirely by the second timepoint, compared to only 8% in the control (Fig. 6C). This suggests that the increase in EV shedding observed in the *inpp-1* mutant might result from all of the PKD-2::GFP in a subset of RnB cilia being released in EVs. Excluding cilia that lacked PKD-2::GFP, loss of *inpp-1* caused a significant increase in PKD-2::GFP fluorescence intensity, without impacting PKD-2 volume in the cilium proper (Fig. 6D,E). While analyzing these images, we noticed that loss of *inpp-1* also caused a significant backup of PKD-2::GFP in the RnB dendrites (Fig. 6F–H). Together, these results indicate that altered phosphoinositide metabolism in the *inpp-1* mutant affects both the abundance and distribution of PKD-2 in RnB dendrites and cilia.

## INPP-1 and INPP-5k have both overlapping and divergent functions

Phylogenetic analysis shows that the INPP-5k phosphoinositide 5-phosphatase is the closest paralog to INPP-1 in *C. elegans* with high similarity between the catalytic domains of these phosphatases

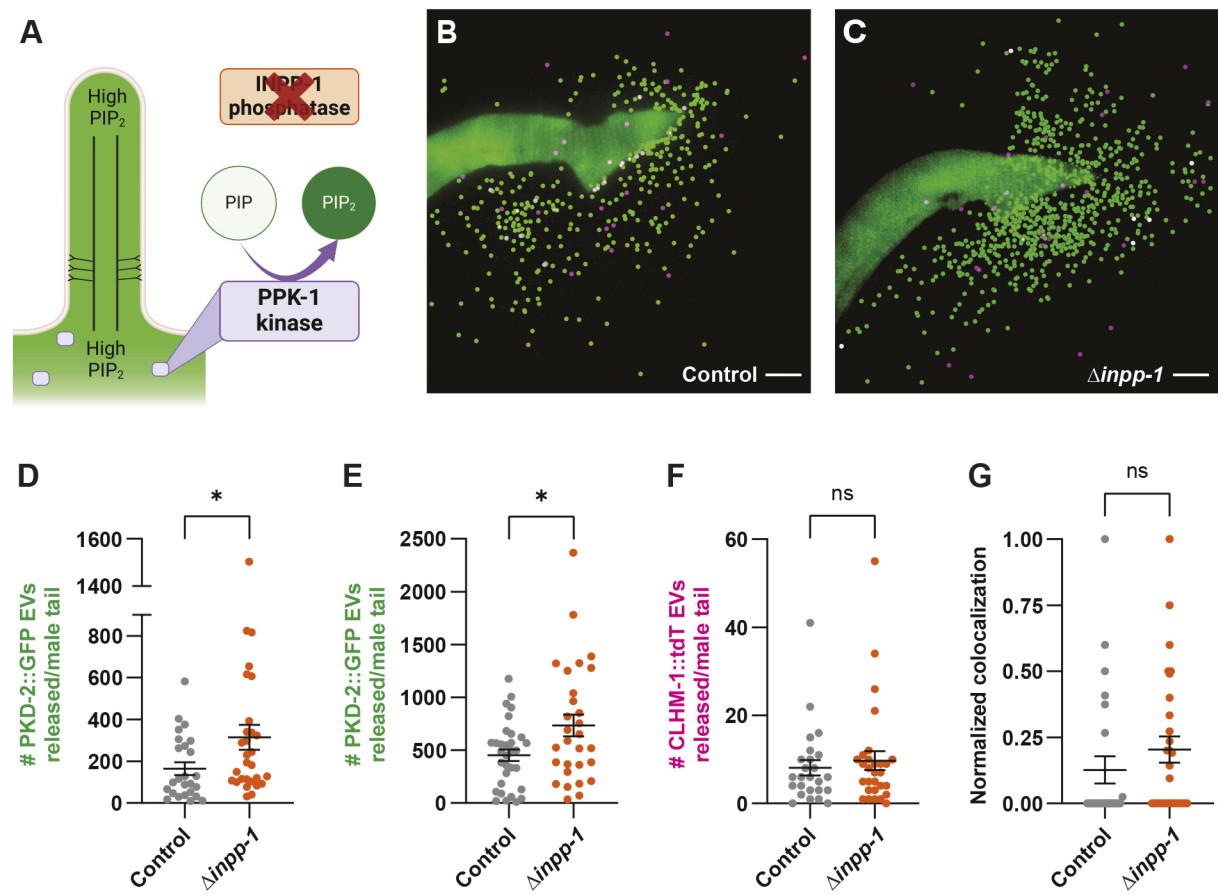

**Fig. 5. Loss of INPP-1 increases the release of PKD-2 ectosomes.** (A) Schematic showing high PI(4,5)P$_2$ in both the ciliary base and cilium proper in the *inpp-1* mutant. (B,C) Representative images of CLHM-1::tdT and PKD-2::GFP EVs shed from (B) control and (C) *inpp-1* mutant male tails. Scale bars: 10 μm. EVs marked with spots (Imaris software) for visualization; see also Fig. S4. (D,E) Loss of *inpp-1* causes an increase in the number of PKD-2::GFP EVs released into the environment. Two different strains were analyzed, (D) UDE275 [*henSi3*; *inpp-1(gk3262)*; *henSi21 him-5(e1490)*] and (E) UDE340 [*mks-2(syb7299)*; *inpp-1(gk3262)*; *henSi21 him-5(e1490)*], to demonstrate reproducibility in independent genetic backgrounds. *n*≥25 in D; *n*≥29 in E. (F,G) CLHM-1::tdT EV shedding (F) and colocalization of the two cargoes in EVs (G) was not altered in the *inpp-1* deletion mutant; *n*≥25. Data are represented as mean±s.e.m. *P<0.05; ns, not significant (two-tailed Mann–Whitney test).

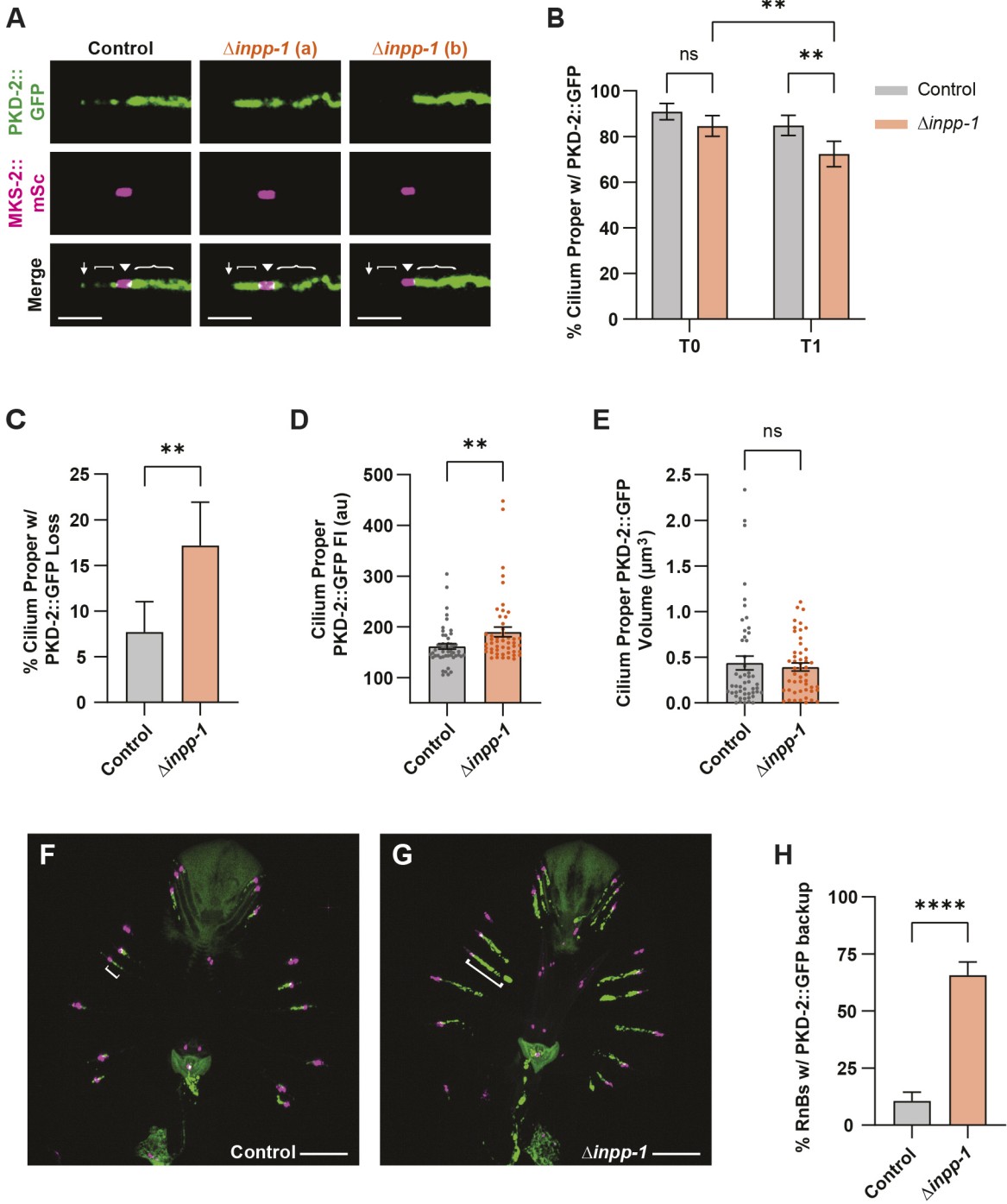

**Fig. 6. INPP-1 regulates localization of PKD-2 in the RnB neurons.** (A) PKD-2::GFP (top) and MKS-2::mSc (middle) in cilia of control and *inpp-1* mutant (two representative cilia; a,b) animals. Identical settings used for all images; cilium distal tip (↓) is oriented to the left; cilium proper ([), transition zone (▼) and ciliary base ({) are labeled. Scale bars: 2 μm. (B) Percentage of *inpp-1* mutant and control RnB neurons that exhibited PKD-2::GFP in the cilium proper at 5 min (T0) and 45 min (T1) after being prepared for imaging; *n*≥65. There was no difference at the initial time point, but significantly more *inpp-1* mutant cilia lacked PKD-2::GFP at the later time point. (C) Analysis of only the RnBs with PKD-2::GFP present in the cilium at the first time point that still showed GFP by 45 min; the *inpp-1* mutant had significantly fewer RnB cilia with PKD-2::GFP. *n*≥55. (D,E) Mean fluorescence intensity (FI) (D) and volume (E) of PKD-2::GFP in the cilium proper of control and *inpp-1* mutants. Cilia that lacked PKD-2::GFP signal in the cilium proper detectable by the Imaris surfaces tool at a threshold of 16 were excluded from this analysis, *n*=50. (F,G) Representative images of PKD-2::GFP (green) and MKS-2::mSc (magenta) in the RnB neurons of control (F) and *inpp-1* mutant (G) male tails; bracket shows PKD-2::GFP in the ciliary base and dendrite of R4B. Scale bars: 10 μm. (H) Loss of *inpp-1* caused a significant increase in the number of RnB neurons with PKD-2 backed-up in the dendrites, defined as the percentage of neurons with PKD-2::GFP signal extending greater than 5 μm from the TZ; *n*≥66. Data are represented as mean±s.e.m. *P<0.05, **P<0.01, ****P<0.0001; ns, not significant [binomial test (B,C,H); two-tailed Mann–Whitney test (D,E)]. au, arbitrary units; w/, with.

(Bae et al., 2009). A mutation in *inpp-5k*, previously known by the gene name *cil-1*, was originally identified in a genetic screen for mutants with PKD-2 localization defects in the EVNs (Bae et al., 2008). In *inpp-5k* mutants, there is an increase in PKD-2::GFP in the dendrites, axons and cell bodies, whereas PKD-2 localization in the cilium proper appears unchanged (Bae et al., 2009). We crossed

the *inpp-5k(my15)* early stop mutation with a PKD-2::GFP single-copy transgene and MKS-2::mSc TZ marker. Loss of *inpp-5k* caused back-up of PKD-2::GFP signal in 71% of RnB dendrites, compared to only 26% of dendrites in the control (Fig. 7A–C). This confirms the effect of *inpp-5k(my15)* on PKD-2 localization, which had previously been observed with a PKD-2::GFP overexpression transgene.

Given that the *inpp-1* mutant and *inpp-5k* mutant both exhibited aberrant PKD-2 localization in the RnB dendrites, we hypothesized that these phosphatases could have redundant functions in the regulation of PI(4,5)P$_2$ level in the cilium proper, ciliary abundance of the EV cargoes and EV shedding. First, to determine whether INPP-5k and INPP-1 exhibit a similar localization pattern, we

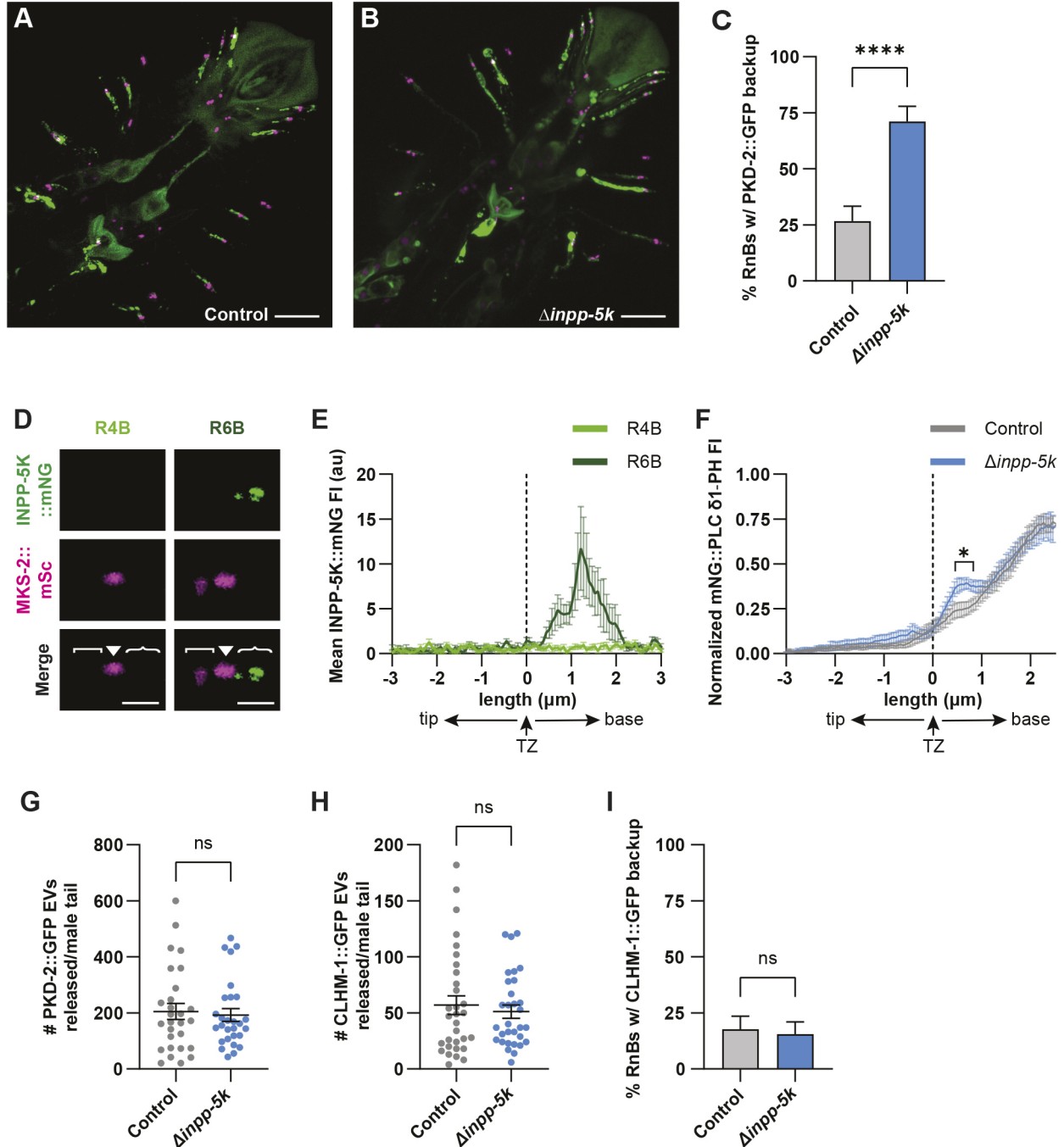

**Fig. 7. Loss of *inpp-5k* impacts PKD-2 localization in neurons, but not ectosome release.** (A,B) Representative images of PKD-2::GFP (green; *henSi20*) and MKS-2::mSc (magenta) in the RnB neurons of (A) control and (B) *inpp-5k* mutant male tails. Scale bars: 10 µm. (C) Loss of *inpp-5k* causes backup of PKD-2::GFP in RnB dendrites. *n*≥45. (D) INPP-5K::mNG (top) and MKS-2::mSc (middle) in R4B (left) and R6B (right) neurons. In these images, the cilium proper ([), transition zone (▼) and ciliary base ({) are indicated. Scale bars: 2 µm. (E) Average INPP-5K::mNG fluorescence intensity (FI) shows that INPP-5K is present in the ciliary base of R6B (dark green), but absent from R4B (light green). Dotted line indicates the center of the TZ; *n*≥8 for each RnB cilia. (F) mNG::PLCδ1-PH fluorescence intensity in control (gray) and *inpp-5k* mutant (blue) animals; *n*≥13 R3B–R5B cilia analyzed. A small increase in PI(4,5)P$_2$ in the PCMC is observed in the *inpp-5k* mutant; *P*<0.05 for all points indicated. (G–I) Loss of *inpp-5k* does not impact release of (G) PKD-2::GFP EVs (*n*≥28), (H) CLHM-1::GFP EVs (*n*≥31) or (I) distribution of CLHM-1::GFP in the RnB dendrites (*n*≥45 cilia). Data are represented as mean±s.e.m. *P<0.05, ****P<0.0001; ns, not significant [binomial test (C,I); two-tailed Mann–Whitney test (F–H)]. au, arbitrary units; w/, with.

created an INPP-5k::mNG endogenous reporter. *inpp-5k* is expressed in an operon, and previous transgenes utilized to look at INPP-5k localization in the EVNs have used the *pkd-2* regulatory sequences (Bae et al., 2009). INPP-5k::mNG retains partial function as animals expressing this endogenous reporter had significantly higher brood size compared to *inpp-5k(my15)* mutants, but significantly reduced fertility compared to wild type (Fig. S6A,B). We observed distinct localization of INPP-5k::mNG to the ciliary base of R6B (Fig. 7D,E) as well as the PHA, PHB and PHC ciliated neurons in the male tail (data not shown). However, we did not detect INPP-5k::mNG in the ciliary base of RnBs 1–5 and 7–9, which directly contact and release EVs into the external environment (Fig. 7D,E). Furthermore, INPP-5k::mNG was not observed in the cilium proper of any RnB neurons (Fig. 7D,E). Considering that the partial function of the INPP-5k::mNG reporter could impact its localization and that loss of *inpp-5k* does cause a phenotype in the RnB dendrites (Fig. 7A–C), we proceeded to investigate PI(4,5)P$_2$ in the *inpp-5k* mutant. Loss of *inpp-5k* resulted in a 14% increase in mNG::PLCδ1-PH intensity in the PCMC, just adjacent to the TZ, but had no effect on fluorescence intensity in the cilium (Fig. 7F). This indicates that INPP-1 and INPP-5k do not function redundantly in the cilium proper to regulate PI(4,5)P$_2$ levels.

We next sought to determine whether the dendritic accumulation of PKD-2::GFP observed in the phosphoinositide 5-phosphatase mutants could impact PKD-2 EV release. We found that loss of *inpp-5k* did not alter PKD-2 EV shedding (Fig. 7G), suggesting that the dendritic backup of PKD-2 does not underlie the increase in EVs observed in the *inpp-1* mutant (Fig. 5D,E). These results also show that the different phosphoinositide 5-phosphatases have divergent functions as INPP-1, but not INPP-5k, regulates EV release. Compared to the control, loss of *inpp-5k* did not change the shedding of CLHM-1 EVs or the distribution of CLHM-1 in RnB dendrites (Fig. 7H,I). Thus, altered phosphoinositide metabolism in the *inpp-5k* mutant only influences dendrite enrichment of PKD-2, but not the CLHM-1 ion channel.

## An increase in ciliary PI(4,5)P$_2$ does not impact EVN cilium length

PI(4,5)P$_2$ in the cilium proper has been shown to impact cilium length by regulating membrane stability and turnover (Jacoby et al., 2009; Stilling et al., 2022; Ukhanov et al., 2022). Highly elevated PI(4,5)P$_2$ in the cilium proper results in excision of the cilium distal tip, leading to loss of the cilium in a process known as decapitation (Phua et al., 2017). To determine whether altered PI(4,5)P$_2$ abundance impacts the length of EVN cilia, we used a transgene expressing the KLP-6 kinesin-3 fused to GFP to completely fill out the RnB neurons and labeled the TZ with MKS-2::mSc, then measured cilia in PPK-1 overexpressor, *inpp-1* mutant and control animals (Fig. S2C). Changing PI(4,5)P$_2$ levels did not significantly alter the length or diameter of the cilium proper for three different classes of RnB neurons analyzed, although the highest variability was observed in the *inpp-1* deletion mutant (Fig. 8A–C; Fig. S7A,B). Furthermore, these cilia retain sensory function, as *inpp-1* mutants do not exhibit any defects in mating behavior (Table S1). These data show that an increase in PI(4,5)P$_2$ in the cilium proper does not cause ciliary decapitation or shortening of the EVN cilia.

## DISCUSSION

The phosphoinositide PI(4,5)P$_2$ plays crucial roles in protein localization, endocytosis and membrane deformation (Katan and Cockcroft, 2020). Here, using fluorescently tagged cargoes to visualize EVs, we show for the first time that PI(4,5)P$_2$ also affects the shedding of signaling ectosomes. High PI(4,5)P$_2$ levels differentially impacted two distinct EV subpopulations, decreasing biogenesis of CLHM-1-containing ectosomes derived from the ciliary base, but increasing budding of PKD-2 EVs from the cilium distal tip (Fig. 8D). Although reduced CLHM-1 EV shedding in the PPK-1 overexpression animals was associated with lower abundance of the EV cargo in the ciliary base, a decrease in PPK-1 function enhanced CLHM-1 EV shedding without affecting ciliary CLHM-1 levels, suggesting that PI(4,5)P$_2$ directly regulates ectosome shedding from this compartment. In contrast, an increase in PI(4,5)P$_2$ in the cilium proper led to time-dependent loss of PKD-2 from distinct cilia, suggesting that the increase in EV shedding from the distal tip of *inpp-1* mutants is the result of abundant PKD-2 EV release from a subset of cilia. Overall, increasing PI(4,5)P$_2$ did not impact cilium length, suggesting that PI(4,5)P$_2$-dependent modulation of ectosome shedding has the potential to regulate EV-mediated signaling without disrupting cilium structure.

We postulate that actin polymerization, which is stimulated by high PI(4,5)P$_2$ in primary cilia (Phua et al., 2017) and triggers release of tip-derived ectosomes (Nager et al., 2017), causes the enhanced EV release observed in the *inpp-1* mutant. Interestingly, loss of *inpp-1* caused a significant increase in the number of RnB neurons that started with, but then lost all, PKD-2::GFP signal in the cilium proper within the timeframe that we analyze EV release. This suggests that heightened EV shedding from a subset of RnB cilia underlies the increase in the number of EVs released in the *inpp-1* mutant. Even with time-lapse imaging, it is not possible to visualize which specific EVs come from each of the individual EV-releasing cilia that express PKD-2 in the male tail. However, to support the idea that altered EV shedding is due to changes in cilium PKD-2, rather than dendritic backup of this protein in the *inpp-1* mutant, we note that aberrant accumulation of PKD-2 in the dendrites of the *inpp-5k* mutant did not affect EV release. This is consistent with the observation that changes in ectocytosis and protein accumulation within a specific ciliary compartment often, although not always, appear to be associated (Clupper et al., 2022; Lobo et al., 2025 preprint; Razzauti and Laurent, 2021; Wang et al., 2021).

Overexpression of PPK-1 decreased CLHM-1 abundance in the ciliary base and EV shedding from this compartment. Whether reduced release of CLHM-1 EVs in the PPK-1 overexpressor is the result of PI(4,5)P$_2$ directly modulating EV biogenesis or due to the changes in cargo enrichment is unclear. Ciliary membrane proteins are internalized at the ciliary base via endocytosis (Clement et al., 2013; Pedersen et al., 2016), a PI(4,5)P$_2$-dependent process (De Craene et al., 2017; Posor et al., 2015). CAV-1, the ortholog of caveolin-1, and DYN-1, the ortholog of dynamin, both localize to the ciliary base and play important roles in endocytosis (Kaplan et al., 2012; Scheidel et al., 2018). Recently, loss of *cav-1* as well as overexpression of dominant-negative DYN-1(K46A), which reduces endocytosis in the ciliary base, has been shown to increase release of ciliary EVs containing the tetraspanin TSP-6 (Lobo et al., 2025 preprint). Thus, it is possible that higher PI(4,5)P$_2$ in the ciliary base of the PPK-1 overexpressor causes an increase in CLHM-1 endocytosis, resulting in a decrease in ectocytosis of EVs carrying this cargo. This could be resolved in future studies by altering PI(4,5)P$_2$ in the ciliary base of endocytosis mutants, and then determining the impact on CLHM-1 EV shedding. Nevertheless, our results showing that reduced PPK-1 function increases CLHM-1 EV shedding without affecting ciliary abundance of this cargo indicate that PI(4,5)P$_2$ can directly inhibit ectosome release from the ciliary base. As PI(4,5)P$_2$ has opposing effects on ectocytosis from the base compared to the cilium tip, we hypothesize that PI(4,5)P$_2$ acts through different mechanisms to

Journal of Cell Science

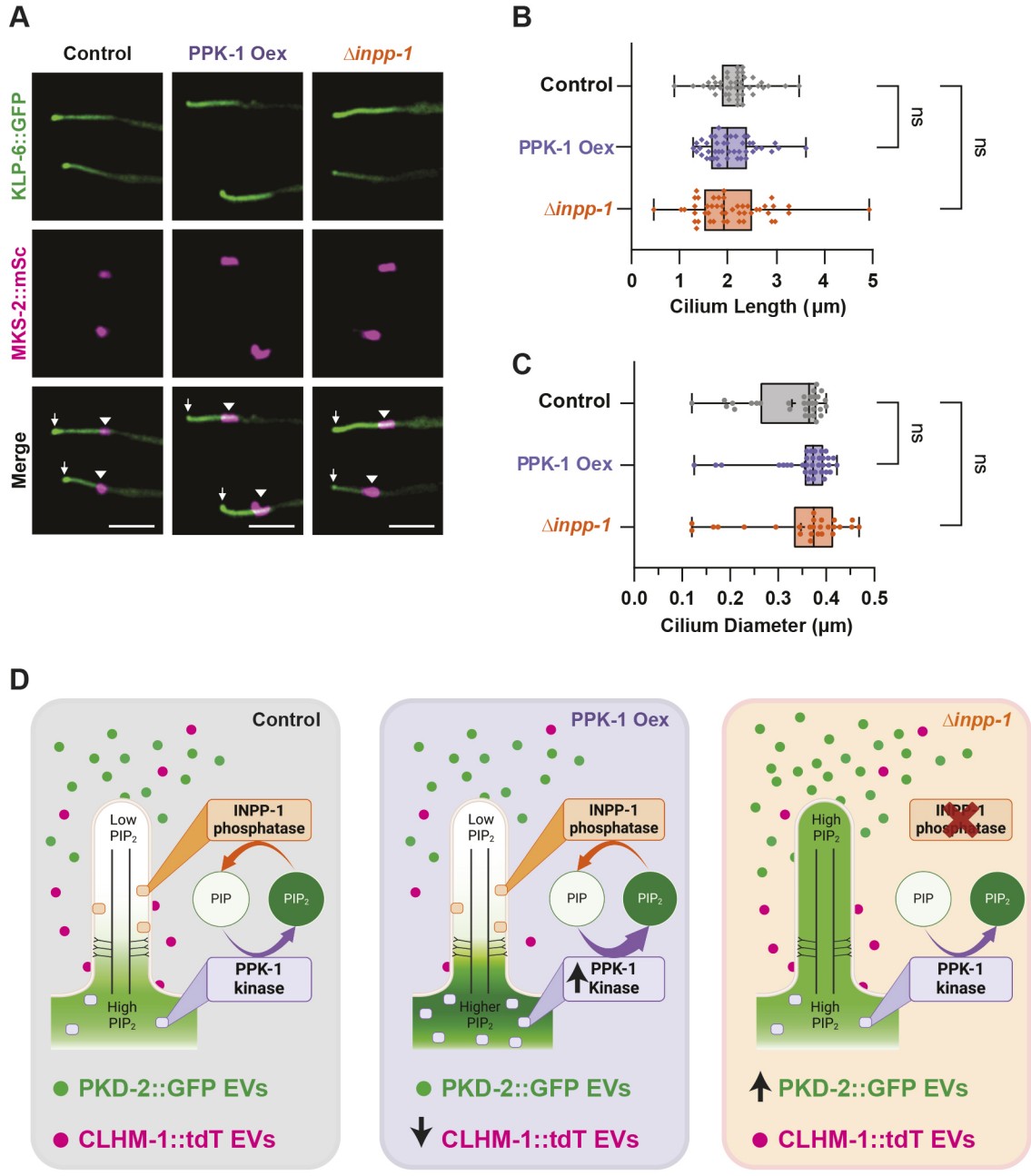

**Fig. 8. An increase in ciliary PI(4,5)P$_2$ does not impact the length of the cilium proper.** (A) Representative images of RnB cilia filled out with KLP-6:: GFP (top) in control (left), PPK-1 overexpression (Oex; middle), and *inpp-1* mutant (right) animals. Cilium distal tip (↓) is to the left; MKS-2::mSc (middle) marks the TZ (▼). Scale bars: 2 μm. (B,C) Length (B) and diameter (C) of cilia were unchanged in the PPK-1 Oex (purple) and *inpp-1* mutant (orange) compared to the control (gray); $n \geq 45$. The box represents the 25–75th percentiles, and the median is indicated. The whiskers show the minimum to maximum range. ns, not significant [mixed-effects model (REML)]. (D) PPK-1 and INPP-1 regulate PI(4,5)P$_2$ abundance and distribution in the RnB ciliated sensory neurons. PI(4,5)P$_2$ is normally absent from the cilium proper (left). Overexpression of PPK-1 (middle) increases PI(4,5)P$_2$ in the PCMC, which leads to a decrease in CLHM-1 abundance and EV shedding, while reduced PPK-1 function causes an increase in CLHM-1 EV release. Loss of *inpp-1* (right) results in accumulation of PI(4,5)P$_2$ in the cilium proper and an increase in PKD-2 EV shedding. Increasing PI(4,5)P$_2$ levels did not impact cilium length.

regulate EV shedding from these compartments. Although the mechanism by which PI(4,5)P$_2$ inhibits EV release from the ciliary base is completely unresolved, it is intriguing that ARF6 and RhoA, two proteins that promote ectosome biogenesis, both activate PIP5K1, the mammalian ortholog of PPK-1 (Honda et al., 1999; Li et al., 2012; Muralidharan-Chari et al., 2009; Weernink et al., 2004). Given that PPK-1-dependent changes in PI(4,5)P$_2$ alter the quantity of EVs released, it is possible that ARF6 and RhoA not only mediate EV ectocytosis, but also regulate the overall amount of shedding.

Phosphoinositide metabolism impacts the localization and abundance of ciliary proteins through effects on trafficking between different ciliary and membrane compartments (Bae et al., 2009; DiTirro et al., 2019). Altering expression of PI(4,5)P$_2$ regulatory enzymes changed relative ciliary distribution of PKD-2, possibly by affecting IFT of this protein. Tubby (TUB) and Tubby-like (TULP) proteins rely on PI(4,5)P$_2$ to link ciliary membrane proteins to the IFT system in the PCMC, while low PI(4,5)P$_2$ in the cilium proper is thought to weaken this interaction and allow for cargo release

(Chávez et al., 2015; DiTirro et al., 2019; Garcia-Gonzalo et al., 2015; Mukhopadhyay et al., 2010, 2013). Nevertheless, high $PI(4,5)P_2$ in the cilium proper does not explain the dendritic accumulation of PKD-2 in the *inpp-1* mutant. Phosphoinositides are also required to direct the trafficking of proteins between endocytic compartments (Bae et al., 2009). Thus, overall disruption of the balance of phosphoinositide species in the *inpp-1* and *inpp-5k* mutants could cause defects in PKD-2 endocytic trafficking, leading to aberrant dendritic accumulation. Interestingly, loss of *inpp-5k* did not increase CLHM-1 abundance in RnB dendrites. The localization of other ciliary proteins including the IFT components OSM-6 and BBS-5, the TRPV ion channel OSM-9 and the G-protein-coupled receptor ODR-10 has also been found to be unchanged in the *inpp-5k* mutant (Bae et al., 2009). Together, this shows that phosphoinositide metabolism is required for proper distribution and degradation of some, but not all ciliary proteins.

How compartmentalization of phosphoinositides in the cilium is established and maintained is not completely understood. Here, we show that PPK-1 and INPP-1 localize to the ciliary base and cilium proper, respectively. Loss of *inpp-1* increased $PI(4,5)P_2$ in the cilium proper, whereas PPK-1 overexpression did not, likely due to the intact TZ. Loss of the TZ protein MKS-5 results in abnormal $PI(4,5)P_2$ accumulation in the cilium proper (Jensen et al., 2015), although whether this is due to mislocalization of $PI(4,5)P_2$ regulatory enzymes, or because the TZ acts as a lipid diffusion barrier (Park and Leroux, 2022) is unknown. Interestingly, TZ mutants also have defects in EV shedding (Wang et al., 2024a), which could result from loss of $PI(4,5)P_2$ compartmentalization. Future work assessing both the abundance and localization of $PI(4,5)P_2$, INPP-1, and PPK-1 in TZ mutants would shed light on the mechanism by which specific TZ proteins contribute to the ciliary compartmentalization of $PI(4,5)P_2$ to regulate protein enrichment and ectosome release.

Loss of *Inpp5e* or rapid synthesis of $PI(4,5)P_2$ in the cilium proper has previously been shown to reduce cilia length through a process known as ciliary decapitation (Jacoby et al., 2009; Phua et al., 2017; Stilling et al., 2022). As a result, INPP5E mutations in humans cause cilium instability and signaling defects, resulting in ciliopathy syndromes (Bielas et al., 2009; Jacoby et al., 2009). We found that the increase in cilium $PI(4,5)P_2$ and tip-derived EV shedding in the *inpp-1* mutant did not change RnB neuron cilium length or morphology. We considered that there could be another redundant phosphatase that prevents $PI(4,5)P_2$ from reaching a level that would induce ciliary decapitation. Loss of the other *Inpp5e* ortholog in *C. elegans*, *inpp-5k*, did not alter the $PI(4,5)P_2$ level in the cilium proper or PKD-2 EV release from the distal tip. In contrast, a recent study found that there was increased ciliary EV shedding from the PHA and PHB neurons in the *inpp-5k* mutant (Lobo et al., 2025 preprint). This could be because INPP-5k localizes to the ciliary base in the PHA and PHB neurons and R6B, but was not detected in the RnBs that release abundant PKD-2 EVs from the distal tip and into the environment. There are other phosphoinositide 5-phosphatases that can dephosphorylate $PI(4,5)P_2$, including OCRL, which localizes to primary cilia in other organisms (Conduit et al., 2012; Jacoby et al., 2009; Zhang et al., 1995). Knockdown of OCRL1 in cultured cells and zebrafish causes ciliary defects (Luo et al., 2012; Rbaibi et al., 2012). We tagged *C. elegans* OCRL-1 with mNeonGreen at the endogenous locus, but did not observe localization to the cilium proper in the EVNs (data not shown). The synaptojanin ortholog UNC-26, which is required for synaptic vesicle endocytosis and recycling, also exhibits sequence similarity with INPP-1; however, *unc-26* mutants do not cause defects in ciliary PKD-2 localization (Bae et al., 2009). Although we cannot rule out the presence of

additional phosphoinositide 5-phosphatases in the RnB cilia, we suggest that since these EVNs are terminally differentiated, this increase in $PI(4,5)P_2$ in the primary cilium might regulate ectosome biogenesis rather than induce ciliary decapitation.

$PI(4,5)P_2$ is partitioned in the EVNs, just as observed for other *C. elegans* ciliated sensory neurons that do not exhibit a high rate of EV release. This suggests that a difference in phosphoinositide metabolism in the EVNs does not account for the abundant shedding of EVs from the cilium distal tip and that $PI(4,5)P_2$ plays a modulatory role in regulating ectosome biogenesis. EV shedding can affect protein composition of the cilium and remove activated receptors to control signaling in the releasing cell (Nager et al., 2017), whereas uptake of EVs can impact signaling in surrounding cells (van Niel et al., 2018) and play a role in inter-organism communication (Wang et al., 2014, 2020). Here, we consider how ciliary abundance and compartmentalization of $PI(4,5)P_2$ could be controlled in living animals to regulate EV biogenesis. External stimuli can dynamically regulate protein content in both the cilium proper and EVs (Clupper et al., 2022; DiTirro et al., 2019; Nager et al., 2017; Wang et al., 2021). Interestingly, compromised sensory signaling in the ciliated sensory AWB neurons increases trafficking of PPK-1 into the cilium proper, and thus, $PI(4,5)P_2$ (DiTirro et al., 2019). This raises the possibility that sensing of mate availability, which impacts EV release (Clupper et al., 2022; Wang et al., 2021), or other mating cues (Srinivasan et al., 2008) by the EVNs, could alter the localization, abundance or activity of the proteins that regulate $PI(4,5)P_2$ levels. Although *inpp-1* mutants have a diminished sensory response and olfactory habituation to low concentrations of diacetyl (Larsch et al., 2015), we found that loss of *inpp-1* did not affect male mating behavior. Thus, it is conceivable that altering $PI(4,5)P_2$ in the RnB neurons could change the signaling potential of EV release without affecting male mating competency.

## MATERIALS AND METHODS

### *C. elegans* strains and maintenance

All strains were cultured at 20°C on nematode growth medium (NGM; Davis and Tanis, 2022) plates seeded with OP50 *E. coli. stam-1(ok406)* I, *inpp-5k(my15)* III, *inpp-1(gk3262)* IV and *him-5(e1490)* V mutant alleles, *ppk-1(syb8819 [mNG::ppk-1])* I, *mks-2(oq101 [mks-2::mNG])* II, *mks-2(syb7299 [mks-2::mSc])* II, *inpp-5k(syb9399 [inpp-5k::mNG])* III and *inpp-1(syb3371 [inpp-1::mNG])* IV endogenous insertion alleles, *gqIs25 [rab-3 promoter:: ppk-1]* I, *myIs10 [klp-6 promoter::klp-6::GFP]*, and *henIs1 [klp-6 promoter::mNG::PLCδ1–PH]* V integrated transgenes, and *henSi3 [clhm-1 promoter::clhm-1::tdT]* III, *henSi20 [pkd-2 promoter::pkd-2::GFP]* IV, and *henSi21 [pkd-2 promoter::pkd-2::GFP]* V single-copy transgenes were utilized. Duplex genotyping was used to detect *stam-1(ok406)* and *inpp-1(gk3262)*; SuperSelective genotyping was used to detect *inpp-5k(my15)* (Touroutine and Tanis, 2020). The presence of fluorescent transgenes was determined by visual screening. See Table S2 for a list of all strains used in this work. Some of these strains are available from the *Caenorhabditis* Genetics Center (CGC); all others are available upon request.

### Generation of the $PI(4,5)P_2$ sensor

The construct used to express the $PI(4,5)P_2$ sensor mNG::PLCδ1-PH in the EVNs was created using restriction enzymes for cloning. mNG containing synthetic introns, with no stop codon, and followed by the flexible linker GSSGSSGTS was amplified from Addgene plasmid #177338 pGLOW77 (Witten et al., 2023) and inserted into pENM1, a plasmid which contains the 1.58 kb *klp-6* promoter (Clupper et al., 2022). The PH domain of phospholipase C δ1 (Stauffer et al., 1998) was amplified from Addgene plasmid #21179 (GFP-C1-PLCdelta-PH) and inserted following the flexible linker at the C-terminal end of mNG. This *klp-6* promoter:: mNG::PLCδ1-PH plasmid pJT174 (2 ng/μl), pCFJ421 (*myo-2* promoter::GFP::H2B; 5 ng/μl) (Frøkjær-Jensen et al., 2012) and genomic DNA (90 ng/μl) were

injected into wild-type animals using the standard germline transformation technique to generate a complex array (Kelly et al., 1997). We screened five lines and chose to proceed with the strain that had the lowest expression and did not disrupt the morphology of the RnB neurons. Part of the resulting extrachromosomal array was integrated into the *him-5* locus using CRISPR/Cas9 genome editing. Two gRNAs specific for the *him-5* locus and one gRNA to target the pJT174 backbone to fragment the extrachromosomal array were injected along with a *dpy-10* cRNA and *dpy-10* repair oligonucleotide harboring the dominant *cn64* mutation. In the F1 generation, animals with the dominant Roller phenotype were cloned to enrich for animals with CRISPR/Cas9 edits (Arribere et al., 2014). We screened resulting F2 progeny and selected four plates that appeared to have significantly more animals with GFP in the nuclei of the pharyngeal muscles (pCFJ421) compared to the starting strain with the extrachromosomal array. Eight worms were cloned from each of these plates and the F3 animals were screened for fluorescence. One plate had animals that exhibited 100% fluorescence and a *him* phenotype; this integrated transgene is designated *henIs1*.

### RnB neuron and cilia imaging and analysis
Imaging of RnB neurons was performed on adult males at 24 h post fourth larval (L4) stage. *C. elegans* were immobilized with 50 mM levamisole (Thermo Fisher Scientific, cat. #AC187870100) pipetted onto 3% agarose pads on microscope slides, and *Z*-stack images of splayed male tails were acquired with an Andor Dragonfly microscope (63× objective) and Zyla sCMOS camera. Identical image acquisition settings were used for all images that were directly compared; control and experimental strains were always imaged on the same day. The 'multi plot' function in ImageJ (NIH) was used to plot fluorescence intensity distribution of mNG::PLCδ1–PH, mNG::PPK-1, INPP-1::mNG, INPP-5k::mNG, CLHM-1::tdT and PKD-2::GFP along a linear region of interest (ROI) that was drawn through the entirety of the cilium and base for R3B–R5B neurons (Fig. S2A) using a maximum intensity projection. For each measurement, the center of the TZ as determined by MKS-2::mSc fluorescence was positioned at 0 μm. The average TZ length in the representative images shown in all figures was 1.09±0.06 μm (mean±s.e.m.), but variability in some images resulted in a spread in MKS-2::mSc signal as observed in Fig. 1. In normalized intensity plots, all values were normalized to the maximum intensity value in the line scan ROI. RnBs with aberrant PKD-2 accumulation in the dendrites were defined as those with PKD-2::GFP signal extending greater than 5 μm from the TZ. Binary scoring was used to classify each RnB cilium as either having PKD-2 present (score=1) or absent (score=0), if no PKD-2::GFP signal was detected using a threshold of 16 in Imaris (Oxford Instruments).

Imaris was used for quantitative volumetric and fluorescence intensity analysis of the cilium proper and cilium base in three-dimensional reconstructions (Fig. S2B). A ROI was drawn around the ciliary region of each RnB analyzed and a constant pre-set intensity threshold, which was dependent on the fluorescent protein being analyzed, was used to select the fluorescence signal and create a corresponding surface using the surfaces tool for the cilium proper and ciliary base. Cilia without PKD-2 were defined as those that lacked detectable PKD-2::GFP signal in the cilium proper using the Imaris surfaces tool at a threshold of 16. For ciliary length and diameter analysis, surfaces were first made to encapsulate the TZ. A ROI was then drawn to exclude the TZ surface but enclose the KLP-6::GFP-labeled cilium proper. The filament tool was used to draw the cilium structure in 3D space given a set segment seed point threshold (=160); lengths of each filament were recorded (Fig. S2C).

### EV imaging and analysis
All transgenes used to quantify EV shedding were integrated into the genome at single copy, as overexpression can impact EV cargo selection and release (Razzauti and Laurent, 2021). Our *C. elegans* strains have been acknowledged as an *in vivo* model in which to analyze EVs in the most recent MISEV position paper (Welsh et al., 2024). Eight transgenic L4 hermaphrodites carrying the *him-5(e1490)* mutation and single-copy transgenes expressing fluorescent protein-tagged EV cargoes were picked onto 6 cm NGM plates and allowed to grow for 4 days, resulting in a mixed population of adult males and hermaphrodites. Adult males were picked from an unseeded region of the NGM plate to prevent contamination from the *E. coli* food source, then picked

into 20 mM levamisole (100 mM diluted in IMage-iT FX Signal Enhancer medium; Thermo Fisher Scientific, item no. I36933) on 3% agarose pads on slides and covered with high-performance cover glass (Zeiss, item no. 474030-9020-000). To prevent contaminating signal from dust and other particles, before use, the high-performance cover glass was placed in a coverslip holder, cleaned with 200-proof ethanol for 30 min (Thermo Fisher Scientific, item no. 04-355-451), rinsed three times with HPLC H₂O shaking for 10 min (Thermo Fisher Scientific, item no. W5-4), then covered with foil and dried on a heat block. EV images were collected with an Andor Dragonfly microscope and Andor Zyla sCMOS detector using total internal reflection fluorescence (TIRF) microscopy. The TIRF angle of incidence was manually adjusted for each animal to achieve critical angle. All images of released EVs were taken at 30 min (±5 min) after each animal was immobilized on the agar pad; control and experimental strains were always imaged on the same day. Imaris software was used to quantify the number of EVs released into the environment from each male tail. EVs were identified using the 'Spot' function, setting object size to 0.350 μm in diameter. A quality threshold was set for each dataset, as determined by analysis of negative controls. 'Hot' pixels were manually removed. EVs that contained both PKD-2 and CLHM-1 were identified as GFP and RFP spots with a maximum distance of 0.3 μm; these data are presented as the probability of PKD-2::GFP being found in a CLHM-1::tdT EV.

### Brood size and embryonic viability assays
Analysis of brood size and embryonic viability was conducted as described previously (Kwah and Jaramillo-Lambert, 2023). The brood size for each worm was determined as the number of unhatched embryos plus the number of live progeny. Embryonic viability was calculated as the number of live progeny divided by the total number of progeny, multiplied by 100.

### Analysis of male mating behavior
The day before recording male mating behaviors, 3 cm low peptone (0.4 g/l) NGM plates were seeded with 2 drops of 2.5 μl freshly cultured OP50. After 24 h, ten young adult hermaphrodites were picked onto the mating plates and allowed to settle on the bacterial lawns before introducing a single age-synchronized young male worm. The male was recorded for 15 min with an AmScope (MU1000) camera and Amlite software. Male behaviors, including ventral touch time (min), vulva touch time (min) and the number of turns were determined through frame-by-frame analysis of the recordings.

### Statistical analysis
All statistical analyses and graphing were performed using GraphPad Prism version 10. Dataset normality was determined using the Anderson–Darling normality test. A Mann–Whitney $U$ test was used when comparing two datasets and a Kruskal–Wallis test with Dunn's multiple comparisons when comparing three datasets; a bionomial test was used to analyze binary data (*$P<0.05$, **$P<0.01$, ***$P<0.001$, ****$P<0.0001$). Statistical details for individual experiments are specified in the figure legends.

#### Acknowledgements
We thank the *Caenorhabditis* Genetics Center, which is supported by the NIH-ORIP (P40 OD010440), for strains. The authors also thank the University of Delaware BioImaging facility for their support, which was essential for this work. Microscopy access was supported by National Institutes of Health NIGMS P20 GM103446, NIH-NIGMS P20 GM139760, and the State of Delaware; microscopy equipment was acquired with NIGMS S10 OD030321.

#### Competing interests
The authors declare no competing or financial interests.

#### Author contributions
Conceptualization: M.W.E., A.E.S., J.E.T.; Formal analysis: M.W.E., A.E.S., K.D.P., N.S.P., T.K.; Funding acquisition: M.W.E., J.E.T.; Investigation: M.W.E., A.E.S., K.D.P., N.S.P., T.K.; Methodology: M.W.E., A.E.S., J.E.T.; Supervision: J.E.T.; Visualization: M.W.E., A.E.S., J.E.T.; Writing – original draft: M.W.E., A.E.S., J.E.T.; Writing – review & editing: M.W.E., A.E.S., K.D.P., N.S.P., T.K., J.E.T.

#### Funding
This work was supported by the National Institutes of Health NIGMS T32 GM133395 to M.W.E. as part of the Chemistry Biology Interface predoctoral training program, NIGMS R01 GM135433 to J.E.T., and Common Fund R03 TR004480 to J.E.T.

 Deposited in PMC for immediate release.

### Data and resource availability
Original data have been deposited at Mendeley Data (doi:10.17632/t798b8yjnn.1) and are available as of the date of publication. Individual image files as well as any additional information required to reanalyze the data are available from the corresponding author upon request. All other relevant data and details of resources can be found within the article and its supplementary information.

### Peer review history
The peer review history is available online at https://journals.biologists.com/jcs/lookup/doi/10.1242/jcs.264005.reviewer-comments.pdf

### Special Issue
This article is part of the Special Issue 'Cilia and Flagella: from Basic Biology to Disease', guest edited by Pleasantine Mill and Lotte Pedersen. See related articles at https://journals.biologists.com/jcs/issue/138/20.

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
