## [Peer Review File · Journal of Cell Science]

Phosphatidylinositol 4,5-bisphosphate impacts extracellular vesicle shedding from *C. elegans* ciliated sensory neurons

Malek W. Elsayyid, Alexis E. Semmel, Krisha D. Parekh, Nahin Siara Prova, Tao Ke and Jessica E. Tanis

DOI: 10.1242/jcs.264005

Editor: Lotte Pedersen

Review timeline

Original submission:	13 March 2025
Editorial decision:	14 April 2025
First revision received:	20 August 2025
Editorial decision:	12 September 2025
Second revision received:	26 September 2025
Accepted:	26 September 2025

Original submission

First decision letter

MS ID#: jcs.264005

MS TITLE: Phosphatidylinositol 4,5-bisphosphate Impacts Extracellular Vesicle Shedding from *C. elegans* Ciliated Sensory Neurons

AUTHORS: Malek Elsayyid; Alexis Semmel; Nahin Prova; Krisha Parekh; Jessica Tanis

ARTICLE TYPE: Research Article

Dear Dr Tanis,

We have now reached a decision on the above manuscript.

To see the reviewers' reports and a copy of this decision letter, please go to:

Reviewer 1

SUMMARY OF THE ADVANCE MADE IN THIS PAPER AND ITS POTENTIAL SIGNIFICANCE TO THE FIELD

This intriguing study explores the role of phosphatidylinositol (4,5)-bisphosphate (PI(4,5)P₂) in extracellular vesicle (EV) biogenesis in *C. elegans* neurons. While prior work established PI(4,5)P₂'s involvement in EV release via cilia decapitation, this study reveals that elevated PI(4,5)P₂ levels suppress EV release from the ciliary base while promoting distal tip budding. Furthermore, PI(4,5)P₂ regulates EV cargo trafficking and localization but does not affect cargo sorting into distinct ectosome subpopulations or cilium length. The findings are compelling, and the following suggestions could further strengthen the work:

SUGGESTIONS TO AUTHORS

1. Including another primary cilium marker (e.g., acetylated $\hat{\pm}$ -tubulin) in immunostaining would help validate ciliary structural integrity and improve robustness.

2. Recent studies have successfully used scanning electron microscopy (SEM) to visualize EVs attached to primary cilia. Incorporating such images would provide direct morphological evidence supporting the findings.
3. Did the authors observe any changes in cilia diameter or thickness in response to altered PI(4,5)P₂ levels? If data exists, this could offer additional insights into membrane dynamics.
4. Does PI(4,5)P₂ influence MVB formation or MVB-mediated EV release near the ciliary pocket or surrounding regions? Clarifying this could differentiate between direct ectosome budding and MVB-derived exosome pathways.
5. Labeling the ciliary tip and base in relevant figures would improve readability and help readers interpret spatial EV release patterns.
6. The Figure 3 legend references Supplementary Figure S2, but the unmarked images appear to be in Supplementary Figure S3. This discrepancy should be verified.

Reviewer 2

SUMMARY OF THE ADVANCE MADE IN THIS PAPER AND ITS POTENTIAL SIGNIFICANCE TO THE FIELD

Strengths

By exporting signaling molecules and unwanted proteins from cilia, the ciliary extracellular vesicles (EVs) are involved in intercellular communication and maintenance of ciliary homeostasis. Understanding ciliary EVs biogenesis is important to uncover mechanisms limiting or causing cilia dysfunction. Here, the authors take advantage of two ciliary markers, localized at the base and the tip of the cilium, to study how PI(4,5)P₂ affects ciliary EV production from these distinct subdomains. The use of fluorescent knock-in strains allows for precise characterization of cargo distribution within cilia and their respective packaging into EVs in individual animals. These approaches support the conclusion that PI(4,5)P₂ compartmentalization influences both the localization of cargo within the cilium and their loading into EVs. The experimental procedures are well-conceived and generally convincing.

SUGGESTIONS TO AUTHORS

Major weaknesses:

Overall, the manuscript requires careful proofreading for grammar, clarity, and precision. For instance, in the abstract, "high PI(4,5)P₂ differentially affect EV release" could be rephrased to clarify whether the distribution and/or absolute levels of PI(4,5)P₂ are more relevant. Additionally, "alter cargo trafficking" should be revised to specify that only distribution, not trafficking, were demonstrated. I did not comment on everything as a lot remain to be done on the writing. below are comments on conclusions reached by the authors.

- 1) While the two markers are differentially localized and loaded into EVs, it remains unclear whether PI(4,5)P₂ modulates local EV biogenesis rates or merely affects cargo enrichment within a given compartment where EV release is constant. This distinction is essential to fully interpret the functional consequences of PI(4,5)P₂ redistribution.
- 2) The study focuses on inpp-1, but this is only one of two INPP5E homologs in *C. elegans*. The second INPP5E homolog: cil-1 is briefly dismissed in the discussion, yet Bae et al. report effects on EVNs in cil-1 mutants. Greater homology with mammalian INPP5B does not preclude overlapping functions with inpp-1. The possibility of redundancy or divergence between these phosphatases should be explored in terms of localization, impact on EV release, and cargo distribution before to exclude its contribution.
- 3) The authors conclude that endocytosis of CLHM-1 at the PCMC is regulated by PI(4,5)P₂, yet no direct evidence is provided. This claim would be strengthened by analysis of CLHM-1 levels in endocytosis or degradation pathway mutants, as previously done for PKD-2 (e.g., Hu et al., 2007; Kaplan et al., 2012; Scheidel et al., 2018). I realise this can be a lot of work, so testing key mutants could substantiate the claim, lack of result would just require to avoid such a conclusion.

4) As noted by the authors, sensory signaling can alter PI(4,5)P₂, abundance and distribution in cilia. For instance, Larsh et al. (2015) showed altered olfactory habituation in *inpp-1* mutants. Since both PIP₂ and sensory input can influence EV release, the manuscript would benefit from attempts to distinguish between PI(4,5)P₂-induced sensory deficits and effects of PIP₂ on EV production. What are the effects of mating, or sensory defective mutants on PIP₂ marker abundance and distribution in EVNs? What are the effects of *inpp-1*, *cil-1* and *ppk-1* overexpression on mating behaviour? although it is hard to disentangle fully, this intrication should be discussed.

Minor comments

Line-specific Suggestions:

- * Line 155: A reference is needed for the first sentence.
- * Line 164: Please clarify if the PI(4,5)P₂, sensor is a multicopy transgene.
- * Figure 1: No error bars (SEM) are shown for MKS-2::mSc. As ciliary length varies, the localization relative to the tip must vary. The reported 2 ¼m spread for MKS-2 is surprisingly long.
- * Line 177: The claim that the transgene is functional and does not alter function could be supported by a viability or behavioral assay.
- * Line 192: Without localization of the transgene fused to GFP, we cannot assess whether the phenotype results from ectopic expression or simply increased levels.
- * Line 195 / Figure 2: Is the PI(4,5)P₂, marker significantly enriched in the PCMC in *inpp-1* or *ppk-1* overexpression strains? A scatterplot and statistical analysis would help show inter-animal variability. Which test is used in 2B?
- * Line 218: Missing reference.
- * Line 228: Do *ppk-1* overexpression animals show increased internalization of CLHM-1? what is observed for CLHM-1 in endocytosis mutants? See above.
- * Line 232: It remains unclear whether reduced CHLM-1 in EVs stems from reduced EV release or reduced membrane-associated cargo. Could CHLM-1 abundance in PCMC be correlated with EV release in *ppk-1* OE individuals ?
- * Line 236 / Figure 4E: Statistical analysis is missing to support the sentence. Also, the legend and figure text do not clearly state which compartments are affected.
- * Line 243: The conclusion regarding endocytosis is speculative. Please revise or provide supporting evidence. See above.
- * Figure 5F vs. 3F: Control distributions differ significantly. Why? For Figure 5I/J, given the apparent divergence in distributions between N2 and *inpp-1*, is the Mann-Whitney test appropriate?
- * Line 255: The existence of two subpopulations is evident and should be analyzed separately. Does the 24% lacking PKD-2 in cilia lack it entirely or only at the tip? Is EV release increased in these animals lacking PKD-2? Please correlate PKD-2 amount/localization with EV output to conclude as done in line 261.
- * Line 275: While the conclusion is reasonable, cilia length variability could still reflect EV release dynamics and its effect on cilia tip. Time-lapse imaging might clarify this.
- * Line 290: The claim that PIP₂, does not affect cargo abundance is not accurate given the mixed PKD-2 outcomes (Fig. 5). Rephrase to reflect this heterogeneity.
- * Line 306: Missing references.

Reviewer 3

Elsayyid et al. investigated the mechanisms by which primary cilia release ciliary extracellular vesicles (EVs), using *C. elegans* as a model system. In order to determine whether biogenesis of ciliary ectosomes is regulated by PI(4,5)P₂ as observed for ciliary decapitation, the authors focused on the role of PI(4,5)P₂, specifically by increasing its levels in cilia through either overexpression of the type I phosphatidylinositol 4-phosphate 5-kinase PPK-1 (PIP5K1) or deletion of the phosphoinositide 5-phosphatase *inpp-1* (INPP5E). As a readout, the authors examined how PI(4,5)P₂, modulation influenced the release of ectosomes containing fluorescently tagged cargo proteins.

While ciliary EVs are likely to play a key role in intercellular communication, their analysis should align with the Minimal Information for Studies of Extracellular Vesicles (MISEV2023) guidelines (Welsh et al., 2024, PMID: 38326288). The selective tagging of distinct ciliary EV subpopulations - using the ion channels CLHM-1 and PKD-2, where CLHM-1 is incorporated into EVs released from the periciliary membrane compartment (PCMC) and PKD-2 into EVs predominantly

shed from the distal tip of the cilium - represents a particularly elegant approach. However, the study lacks critical validation steps in the characterization of these EV subpopulations. Notably, the ability to reliably distinguish true EVs from background signal remains uncertain. Although the limited amount of biological material may constrain downstream analyses, it is nevertheless essential to incorporate complementary techniques, as described in the MISEV guidelines (e.g., cryo-EM, flow cytometry, and molecular assays), to ensure that the quantification reflects bona fide EVs rather than nonspecific fluorescent signals.

First revision

Author response to reviewers' comments

We thank the Editor and Reviewers for carefully reviewing our manuscript. We have addressed all of the Reviewers' comments and questions through the addition of **new data** (Figure 2B, Figure 3G-I, Figure 4G-I, Figure 6B,C,F-H, Figure 7A-I, Figure 8C, Sup. Figure 3A-D, Sup. Figure 5A-E, Sup. Figure 6A-B, and Sup. Table 1), modification of the manuscript text, and explanations in this response. Our responses to specific Reviewer comments are below:

Reviewer 1: *This intriguing study explores the role of phosphatidylinositol (4,5)-bisphosphate (PI(4,5)P₂) in extracellular vesicle (EV) biogenesis in C. elegans neurons. While prior work established PI(4,5)P₂'s involvement in EV release via cilia decapitation, this study reveals that elevated PI(4,5)P₂ levels suppress EV release from the ciliary base while promoting distal tip budding. Furthermore, PI(4,5)P₂ regulates EV cargo trafficking and localization but does not affect cargo sorting into distinct ectosome subpopulations or cilium length. The findings are compelling, and the following suggestions could further strengthen the work:*

SUGGESTIONS TO AUTHORS

1. *Including another primary cilium marker (e.g., acetylated α -tubulin) in immunostaining would help validate ciliary structural integrity and improve robustness.*

Response: We performed new experiments to confirm the structural integrity of the RnB cilia in the *inpp-1* mutants. First, we analyzed six parameters of male mating behavior. Loss of *inpp-1* did not cause any defects in male mating (see **new Supplemental Table 1**), indicating that the cilia are not only intact, but also retain sensory function. Second, some of the amphid and phasmid ciliated sensory neurons dye-fill when worms are exposed to the lipophilic dye Dil (1,1'-Dioctadecyl-3,3',3'- tetramethylindocarbocyanine perchlorate). Neurons in mutants that lack the cilium distal tip do not dye fill. While Dil staining cannot be used to study the RnB neurons, the amphid and phasmid neurons in the *inpp-1* mutant and PPK-1 overexpressor were able to take up Dil. The dye filling data further demonstrate cilia integrity in these mutants, however, we have chosen to only include the male mating behavior data in the manuscript because it is more quantitative. If the Reviewer would like us to include images of Dil-stained worms in the supplement, we could include this.

2. *Recent studies have successfully used scanning electron microscopy (SEM) to visualize EVs attached to primary cilia. Incorporating such images would provide direct morphological evidence supporting the findings.*

Response: Unfortunately, it is not possible to visualize the EVs attached to RnB cilia inside the male tail by SEM (see <https://www.wormimage.org/imageList2.php?&page=1> for SEM images of the male tail). TEM has been used to visualize EVs in the lumen adjacent to the EV-releasing CEM cilia which are clustered together in the head (Maguire et al., 2015). However, this approach is unrealistic for study of the EVs associated with the RnBs as it would be incredibly challenging to find the individual RnB cilia in sections of the fan-like structure of the male tail (see Fig. 6F,G; Fig. 7A,B). Instead, we have provided an image of a CLHM-1::tdT EV adjacent to a RnB cilium in **new Supplementary Figure 3**.

We found no significant difference between the diameter of recently shed CLHM-1-containing EVs in the luminal space adjacent to the ciliary base and the CLHM-1 EVs in the environment (Sup. Fig. S3C,D).

3. *Did the authors observe any changes in cilia diameter or thickness in response to altered PI(4,5)P₂ levels? If data exists, this could offer additional insights into membrane dynamics.*

Response: We performed additional analysis and found that the of cilium diameter in the animals with altered PI(4,5)P₂ levels was not significantly different compared to the wild type. These data are presented in **new Figure 8C**.

4. *Does PI(4,5)P₂ influence MVB formation or MVB-mediated EV release near the ciliary pocket or surrounding regions? Clarifying this could differentiate between direct ectosome budding and MVB- derived exosome pathways.*

Response: MVB-mediated EV release from the ciliary base of the EV-releasing neurons in *C. elegans* has not been observed. The CLHM-1-containing EVs, which originate from the ciliary base are likely ectosomes, as we observe budding of individual EVs directly from the plasma membrane at the base of the cilium. We have included images and analysis of newly shed CLHM-1 EVs in **new Supplemental Figure 3**. In addition, we added the following text to the manuscript at line 225: "These EVs are likely ectosomes, as individual EVs appear to bud directly from the plasma membrane (Sup. Fig. S3C,D), multivesicular bodies (MVBs) have not been observed in EVN cilia, and MVB components are not required for shedding (Clupper et al., 2022; Wang et al., 2014; Wang et al., 2024b)."

5. *Labeling the ciliary tip and base in relevant figures would improve readability and help readers interpret spatial EV release patterns.*

Response: Excellent suggestion! All merge images are now labeled as follows: cilium distal tip (↓), cilium proper (⌈), transition zone (▼), ciliary base (⌋), and scale, 2 μm.

6. *The Figure 3 legend references Supplementary Figure S2, but the unmarked images appear to be in Supplementary Figure S3. This discrepancy should be verified.*

Response: We have fixed this typo.

Reviewer 2: *By exporting signaling molecules and unwanted proteins from cilia, the ciliary extracellular vesicles (EVs) are involved in intercellular communication and maintenance of ciliary homeostasis.*

Understanding ciliary EVs biogenesis is important to uncover mechanisms limiting or causing cilia dysfunction. Here, the authors take advantage of two ciliary markers, localized at the base and the tip of the cilium, to study how PI(4,5)P₂ affects ciliary EV production from these distinct subdomains. The use of fluorescent knock-in strains allows for precise characterization of cargo distribution within cilia and their respective packaging into EVs in individual animals. These approaches support the conclusion that PI(4,5)P₂ compartmentalization influences both the localization of cargo within the cilium and their loading into EVs. The experimental procedures are well-conceived and generally convincing.

SUGGESTIONS TO AUTHORS

Major weaknesses:

Overall, the manuscript requires careful proofreading for grammar, clarity, and precision. For instance, in the abstract, "high PI(4,5)P₂ differentially affect EV release" could be rephrased to clarify whether the distribution and/or absolute levels of PI(4,5)P₂ are more relevant. Additionally, "alter cargo trafficking" should be revised to specify that only distribution, not trafficking, were demonstrated. I did not comment on everything as a lot remain to be done on the writing. Below are comments on conclusions reached by the authors.

Response: We agree with the Reviewer that the previous version of the manuscript lacked precision in many places. To address this concern as well as to describe our new data, we have re-written a majority of the text.

1) *While the two markers are differentially localized and loaded into EVs, it remains unclear whether PI(4,5)P₂ modulates local EV biogenesis rates or merely affects cargo enrichment within a given compartment where EV release is constant. This distinction is*

essential to fully interpret the functional consequences of PI(4,5)P₂ redistribution.

Response: We agree that it is unclear if altered PI(4,5)P₂ in the *inpp-1* mutant and PPK-1 overexpressor directly regulates EV biogenesis or whether the altered ciliary abundance of the cargoes indirectly affects EV shedding; we added more to the Discussion (lines 415-454) to address this.

Further, we collected new experimental data that more directly answer this point. We discovered that the addition of mNG to the N-terminus of PPK-1 in the endogenous reporter caused a partial loss of PPK-1 function (**new Fig. 3G,H**), which led to an increase biogenesis of CLHM-1 EVs (**new Fig. 3I**). The ciliary abundance and distribution of CLHM-1::tdT was not affected by reduced PPK-1 function (**new Fig. 4G-I**), suggesting that rather than simply impacting ciliary enrichment of the EV cargoes, PI(4,5)P₂ can more directly inhibit ectocytosis from the ciliary base. These new results are described on lines 239-244 and lines 292-304.

2) *The study focuses on inpp-1, but this is only one of two INPP5E homologs in C. elegans. The second INPP5E homolog: cil-1 is briefly dismissed in the discussion, yet Bae et al. report effects on EVNs in cil-1 mutants. Greater homology with mammalian INPP5B does not preclude overlapping functions with inpp-1. The possibility of redundancy or divergence between these phosphatases should be explored in terms of localization, impact on EV release, and cargo distribution before to exclude its contribution.*

Response: Thankfully we started experiments last year that have allowed us to address this point in entirety as the *cil-1* mutant is very challenging to work with due to its small brood size. Here and throughout the paper, we refer to this gene as *inpp-5k*, which is the currently established gene name on Wormbase; we indicate that that it was previously known as *cil-1* in the text. We have **added new Figure 7 and new Supplemental Figure 6** to show the effect of the *inpp-5k(my15)* mutation on PKD-2 localization in the RnB dendrites (Fig. 6A-C), PI(4,5)P₂ in the ciliary compartments (Fig. 6F), PKD-2 EV release (Fig. 6G), CLHM-1 EV release (Fig. 6H), and CLHM-1 localization in the RnB dendrites (Fig. 6I). We also created and characterized the localization of an INPP-5k::mNG endogenous reporter (Fig. 6D,E; Sup. Fig 6A,B). We discovered that INPP-1 and INPP-5k have both overlapping and divergent functions, which we describe in text on lines 336-379.

3) *The authors conclude that endocytosis of CLHM-1 at the PCMC is regulated by PI(4,5)P₂, yet no direct evidence is provided. This claim would be strengthened by analysis of CLHM-1 levels in endocytosis or degradation pathway mutants, as previously done for PKD-2 (e.g., Hu et al., 2007; Kaplan et al., 2012; Scheidel et al., 2018). I realise this can be a lot of work, so testing key mutants could substantiate the claim, lack of result would just require to avoid such a conclusion.*

Response: To determine whether enhanced degradation of CLHM-1 can lead to reduced EV shedding, we performed experiments with a *stam-1* mutant. Previously, *stam-1* was shown to be expressed in the RnBs and in *stam-1* mutant males, PKD-2::GFP accumulates in early endosomes in the ciliary base and distal dendrite, suggesting that STAM-1 sorts PKD-2 for lysosomal degradation (Hu et al., 2007). However, we found that loss of *stam-1* did not disrupt ciliary CLHM-1, CLHM-1 EV shedding, or even PKD-2 EV shedding (new Supplemental Figure 5, see lines 262-283 in the manuscript). Additional experimentation will be required to identify specific mutants that disrupt the endocytosis and degradation of CLHM-1. As a result, we are currently unable to define the mechanism by which overexpression of PPK-1 results in reduced CLHM-1 in the ciliary base and EV shedding. We also now address this in the discussion, lines 431-442.

4) *As noted by the authors, sensory signaling can alter PI(4,5)P₂ abundance and distribution in cilia. For instance, Larsh et al. (2015) showed altered olfactory habituation in inpp-1 mutants. Since both PIP2 and sensory input can influence EV release, the manuscript would benefit from attempts to distinguish between PI(4,5)P₂-induced sensory deficits and effects of PIP2 on EV production. What are the effects of mating, or sensory defective mutants on PIP2 marker abundance and distribution in EVNs? What are the effects of inpp-1, cil-1 and ppk-1*

overexpression on mating behaviour? although it is hard to disentangle fully, this intrication should be discussed.

Response: We performed mating assays (see **new Supplemental Table 1**) and found that loss of *inpp-1* caused no change in mating behavior. Mating assays were not carried out with the PPK-1 overexpressor animals because PPK-1 overexpression in all neurons causes uncoordinated movement, which impacts analysis of mating. *cil-1* mutants were previously shown to exhibit normal mating behavior (Bae et al. 2009). After we determined that an increase in PI(4,5)P₂ in the cilium proper of the *inpp-1* mutant did not alter mating, we reasoned that even if mating defective mutants were to have a difference in mNG::PLCδ1-PH abundance or distribution that this would most likely not be the underlying cause of their behavioral defects. However, it remains possible that altering PI(4,5)P₂ in the cilium proper could result in more subtle deficits, such as the ability to tune EV shedding in response to the presence of mates. The last paragraph of the discussion addresses this possibility (see lines 510-526).

Minor comments

Line-specific Suggestions:

* *Line 155: A reference is needed for the first sentence.*

Response: Reference added

* *Line 164: Please clarify if the PI(4,5)P₂ sensor is a multicopy transgene.*

Response: The PI(4,5)P₂ sensor is a multicopy transgene (see lines 551-557 in the Materials and Methods). Originally, we generated a SCI of this sensor, but it was too dim for imaging. We then injected the *k1p-6* promoter::mNG::PLCδ1-PH plasmid at low (2 ng/μl) concentration with genomic DNA and analyzed five lines carrying complex extrachromosomal arrays. The line with the lowest mNG::PLCδ1-PH expression did not disrupt RnB neuron morphology and was selected for integration. It is likely that only part of the extrachromosomal array was integrated through our CRISPR/Cas9 genome editing method, as mNG::PLCδ1-PH expression in the integrated strain is lower than what was observed in the starting extrachromosomal strain.

* *Figure 1: No error bars (SEM) are shown for MKS-2::mSc. As ciliary length varies, the localization relative to the tip must vary. The reported 2 μm spread for MKS-2 is surprisingly long.*

Response: We have added error bars to the MKS-2::mSc line scan data in Figure 1. The average TZ length was 1.09 +/- 0.06 μm; we have now included this in the Figure 1 legend. The larger spread observed for the MKS-2::mSc observed in the linescan is likely due to a number of factors including: 1) differences in the z-plane orientation of some cilia that cause the measurement to be less accurate when the image is converted to a maximum intensity projection for the linescan analysis, 2) multiple peaks due to occasional ciliary in addition to TZ mKate fluorescence, and 3) slight animal movement that we did not detect by eye. Beyond Figure 1, we simply use the MKS-2::mSc as a TZ marker to clearly delineate the cilium proper from the ciliary base.

* *Line 177: The claim that the transgene is functional and does not alter function could be supported by a viability or behavioral assay.*

Response: We completed both brood size and embryonic viability assays; see **new Figure 3 G,H**. We discovered that the mNG::PPK-1 endogenous reporter strain exhibited significantly reduced brood size and embryonic viability, suggesting that the addition of mNG to the N-terminus of PPK-1 results in a partial loss of function; null mutations in *ppk-1* cause embryonic lethality. We were then able to use mNG::PPK-1 to determine how loss of PPK-1 affects CLHM-1 EV release (**new Fig. 3I**) and ciliary localization (**new Fig. 4G-I**).

* *Line 192: Without localization of the transgene fused to GFP, we cannot assess whether the phenotype results from ectopic expression or simply increased levels.*

Response: We added the following statement at line 194 to address this “Since PI(4,5)P₂ only increased where the mNG::PPK-1 endogenous reporter localized, this suggests that the effect is due to overexpression of the kinase. However, we cannot rule out the possibility that

overexpression of PPK-1 also causes ectopic expression as it lacks a tag to study localization.”

* *Line 195 / Figure 2: Is the PI(4,5)P₂ marker significantly enriched in the PCMC in *inpp-1* or *ppk-1* overexpression strains? A scatterplot and statistical analysis would help show inter-animal variability. Which test is used in 2B?*

Response: We have performed additional statistical analyses, which show that the PI(4,5)P₂ reporter is significantly enriched in the cilium proper of the *inpp-1* mutant, while fluorescent intensity of the reporter is significantly greater in the ciliary base of the PPK-1 overexpressor (see Fig 2B,C). A one-way ANOVA was used to compare the three data sets.

* *Line 218: Missing reference.*

Response: Reference added

* *Line 228: Do *ppk-1* overexpression animals show increased internalization of CLHM-1? what is observed for CLHM-1 in endocytosis mutants? See above.*

Response: It is unclear whether there is increased internalization of CLHM-1 in the animals overexpressing PPK-1. During the revision of this manuscript, we did investigate the effect of loss of *stam-1* on CLHM-1 EV release and ciliary localization, but found no significant change compared to wild type (see **new Supplemental Figure 5**). We also added a section to the Discussion (lines 429- 440) to address this.

* *Line 232: It remains unclear whether reduced CHLM-1 in EVs stems from reduced EV release or reduced membrane-associated cargo. Could CHLM-1 abundance in PCMC be correlated with EV release in *ppk-1* OE individuals ?*

Response: Possibly; see revised Discussion, lines 428-454.

* *Line 236 / Figure 4E: Statistical analysis is missing to support the sentence. Also, the legend and figure text do not clearly state which compartments are affected.*

Response: We corrected the text to indicate that PPK-1 overexpression altered the distribution (not abundance) of ciliary PKD-2::GFP as there was an increase in PKD-2 in the transition zone of the PPK- 1 overexpressor compared to the wild type (Fig. 4E,F). Statistical analysis is now indicated on the line scan graph in Fig. 4E.

* *Line 243: The conclusion regarding endocytosis is speculative. Please revise or provide supporting evidence. See above.*

Response: We agree that this is speculative and have removed this statement.

* *Figure 5F vs. 3F: Control distributions differ significantly. Why? For Figure 5I/J, given the apparent divergence in distributions between N2 and *inpp-1*, is the Mann-Whitney test appropriate?*

Response: Regarding Figure 5F vs 3F: day to day, we do see variability in the overall amount of EV release from control animals. We understand some (ex. time post immobilization, temperature), but not all factors that can lead to this variability. For this reason, both control and experimental strains must always be imaged on the same day and all images of released EVs are taken 30 ± 5 minutes after each animal was immobilized on the agar pad. The normalized colocalization shown in Fig. 5G (which used to be Fig. 5F), is based on the PKD-2::GFP EV and CLHM-1::tdT EV release data in Fig. 5D and Fig. 5F, respectively. Overall EV release on the days that these experiments were conducted was on the lower side of what we normally see. Since colocalization is determined as the probability of PKD- 2::GFP being in a CLHM-1::tdT EV, we have found that calculating these values when the CLHM-1::tdT denominator is small can change the distribution, but not the overall average colocalization. Regarding the old Figure 5I,J: we no longer present the data this way.

* *Line 255: The existence of two subpopulations is evident and should be analyzed separately. Does the 24% lacking PKD-2 in cilia lack it entirely or only at the tip? Is EV release increased in these animals lacking PKD-2? Please correlate PKD-2 amount/localization with EV output to conclude as done in line 261.*

Response: We performed additional experiments to analyze PKD-2::GFP distribution in RnB neuron cilia at two different timepoints. Binary scoring was used to classify each RnB as either having PKD-2 present in the cilium proper, tip, or both (score = 1) or entirely absent (score =

0). There was no difference in PKD-2 ciliary localization in the *inpp-1* mutant right after mounting the animals for ciliary imaging, but forty minutes later, on a similar time scale to our EV imaging, PKD-2 was absent from the cilium proper and distal tip in 28% of the *inpp-1* mutant RnB cilia examined, significantly more than wild type (**new Figure 6A,B**). Analysis of only the cilia that contained PKD-2 initially showed that 20% in the *inpp-1* mutant lost PKD-2 signal entirely by the second timepoint, compared to only 8% in the control (**new Figure 6C**). We discuss this correlation between PKD-2 in the cilium and EV release in the Results (lines 319-334) and Discussion (lines 415-427).

* *Line 275: While the conclusion is reasonable, cilia length variability could still reflect EV release dynamics and its effect on cilia tip. Time-lapse imaging might clarify this.*

Response: We have added this as a discussion point on line 419.

* *Line 290: The claim that PIP₂ does not affect cargo abundance is not accurate given the mixed PKD-2 outcomes (Fig. 5). Rephrase to reflect this heterogeneity.*

Response: We agree. We have performed additional experiments, which more definitively show this heterogeneity - see **new Figure 6B-D**.

* *Line 306: Missing references.*

Response: References added.

Reviewer 3: *Elsayyid et al. investigated the mechanisms by which primary cilia release ciliary extracellular vesicles (EVs), using C. elegans as a model system. In order to determine whether biogenesis of ciliary ectosomes is regulated by PI(4,5)P₂ as observed for ciliary decapitation, the authors focused on the role of PI(4,5)P₂, specifically by increasing its levels in cilia through either overexpression of the type I phosphatidylinositol 4-phosphate 5-kinase PPK-1 (PIP5K1) or deletion of the phosphoinositide 5-phosphatase inpp-1 (INPP5E). As a readout, the authors examined how PI(4,5)P₂ modulation influenced release of ectosomes containing fluorescently tagged cargo proteins.*

While ciliary EVs are likely to play a key role in intercellular communication, their analysis should align with the Minimal Information for Studies of Extracellular Vesicles (MISEV2023) guidelines (Welsh et al., 2024, PMID: 38326288). The selective tagging of distinct ciliary EV subpopulations—using the ion channels CLHM-1 and PKD-2, where CLHM-1 is incorporated into EVs released from the periciliary membrane compartment (PCMC) and PKD-2 into EVs predominantly shed from the distal tip of the cilium—represents a particularly elegant approach. However, the study lacks critical validation steps in the characterization of these EV subpopulations. Notably, the ability to reliably distinguish true EVs from background signal remains uncertain. Although the limited amount of biological material may constrain downstream analyses, it is nevertheless essential to incorporate complementary techniques, as described in the MISEV guidelines (e.g., cryo-EM, flow cytometry, and molecular assays), to ensure that the quantification reflects bona fide EVs rather than nonspecific fluorescent signals.

Response: We agree that complementary approaches are essential for analysis EV subpopulations. Using lambda spectral imaging followed by linear unmixing, we demonstrated that CLHM-1::GFP and PKD-2::GFP signal in EVs released into the environment is bona fide GFP emission and not background (Clupper et al., 2022). In addition, both CLHM-1 and PKD-2 were identified as EV cargoes in an EV proteomic data set (Nikonorova et al., 2022). We have now further confirmed that these are bona fide EVs by performing additional experiments; data are shown in **new Supplemental Figure 3**. CLHM-1::tdTomato and PKD-2::GFP labeled EVs were identified by Imaris spot detection in TIRF images of males that express the CLHM-1::tdT and PKD-2::GFP single copy transgenes, but not *him-5* control animals (Sup. Fig. 3A,B). We also found no significant difference between the diameter of CLHM-1-containing EVs that were present in the luminal space adjacent to the ciliary base inside the worm and the CLHM-1 EVs in the environment (Sup. Fig. S3C,D). This further supports that CLHM-1 EVs are shed from ciliary base, travel through the lumen, and then are released into environment. We have added a paragraph (lines 206-228), to address these points.

Second decision letter

MS ID#: jcs.264005R1

MS TITLE: Phosphatidylinositol 4,5-bisphosphate Impacts Extracellular Vesicle Shedding from *C. elegans* Ciliated Sensory Neurons

AUTHORS: Malek Elsayyid; Alexis Semmel; Krisha Parekh; Nahin Prova; Tao Ke; Jessica Tanis

ARTICLE TYPE: Research Article

Dear Dr Tanis,

We have now reached a decision on the above manuscript.

To see the reviewers' reports and a copy of this decision letter, please go to:

As you will see, the reviewers gave favourable reports but Reviewer 2 had a few suggestions for improvement that will require amendments to your manuscript. I hope that you will be able to carry these out because I would like to be able to accept your paper.

Second revision

Author response to reviewers' comments

Response to Reviewers

Reviewer 2:

SUGGESTIONS TO AUTHORS

The clarity of the paper is greatly improved in this version, and the authors have addressed all my previous major -and most minor- comments. It is now a very interesting and well-designed article. Nevertheless, I feel that small improvements—mostly regarding the figures—could further help the reader and would not require additional experiments.

1) Figure 2B: This panel shows mNG::PLC δ 1-PH fluorescence in the cilia proper. The same quantification should also be performed for the PCMC. I understand that the physical boundaries of the PCMC are not well defined; however, with the current presentation, it is not possible to conclude whether PIP2 is increased in the PCMC/dendrite of Δ inpp-1. For example, the limits could be defined as the ciliary base area usually covered by CHLM-1::tdT.

We performed this analysis on 3D reconstructions, defining the ciliary base ROI as the 2 μ m region proximal to the TZ. Since the new analysis is easier to interpret than our line scan data, which was generated with maximum intensity projections, we have removed the line scan (old Fig. 2C) and replaced it with the volumetric analysis (new Fig 2D). Figure legend 2 has been updated accordingly.

2) Figure 3I: This panel shows the effect of reduced PPK-1 activity on CLHM-1::tdT EVs. Does the strain used also express PKD-2::GFP? If so, including an analysis of PKD-2::GFP EVs would provide additional evidence that high or low PPK-1 activity does not affect PKD-2::GFP EV release.

We chose to create this strain without PKD-2::GFP in the background because it enabled us to confidently genotype for the presence of mNG::PPK-1, which is very dim. Thus, we are unable to determine if low PPK-1 activity impacts PKD-2::GFP EV release without constructing another strain.

We do note that overexpression of PPK-1 had no impact on shedding of PKD-2 EVs from the distal tip (Fig. 3D).

3) *Figures 4B, 4C: The figure and text should more clearly state whether the effect of PPK-1 overexpression is on the size of the ciliary compartment or on the fluorescence intensity within these compartments. In Figure 4B, the 34% of cilia with no PCMC fluorescence are difficult to distinguish because blue points on blue histograms are not visible. In some PPK-1 overexpression individuals, CLHM-1::tdT fluorescence also seems lost in the cilia proper—are these the same cilia? Additionally, the text refers to the "ciliary base," whereas the figure shows the cilia proper and the PCMC.*

We have edited the text in the Results (line 256) and Figure 4 legend to more clearly indicate that the impact of PPK-1 overexpression is on CLHM-1::tdT fluorescence in the ciliary base. All colors were carefully chosen to be color-blind friendly. In Figure 4B, the purple points showing the 34% of cilia with no CLHM-1::tdT fluorescence in the ciliary base are hard to see not because of the color, but instead, because there are so many points on top of each other. To address this concern, we added a sentence describing the raw data in the Figure 4 legend: "17 out of 50 PPK-1 Oex animals lacked CLHM-1::tdT in the ciliary base; CLHM-1::tdT was present in the base for all control animals (n=41)." Finally, we have updated the figure such that the label is now "Base" instead of "PCMC."

4) *Figures 4G-I: Could you check whether the PCMC volumes of CLHM-1::tdT controls differ between Figures 4C and 4I? It seems that the outliers seen in 4C are not present in 4I.*

Yes, the average PCMC volume of CLHM-1::tdT in the controls in 4C and 4I does differ. We have added the following sentence to the Figure 4 legend to explain this: "Note, CLHM-1::tdT control values in B,C differ from the values in H,I because imaging was conducted over two years apart by separate individuals on different Andor Dragonfly microscopes." This highlights the importance of always imaging the relevant control and experimental strains on the same day

5) *Figure 5: What is the difference between the strains used in Figures 5D and 5E, and why was this comparison performed? This is not explained in the text or legend.*

The experiment shown in Figure 5E was performed to increase rigor and show reproducibility. We have added the following sentence to the Figure 5 legend: "Two different strains were analyzed, (D) UDE275 (*henSi3; inpp-1(gk3262); henSi21 him-5(e1490)*) and (E) UDE340 (*mks-2(syb7299); inpp-1(gk3262); henSi21 him-5(e1490)*), to demonstrate reproducibility in independent genetic backgrounds."

6) *Figure 6: Did you observe variability in inpp-1::mNG expression/localization (Figure 1) between cilium that could explain the cilium-variable effect of $\Delta inpp-1$ on PKD-2 backup accumulation?*

The INPP-1::mNG fusion protein was present in all cilia analyzed and our line-scan analysis of INPP-1::mNG (Fig. 1D) shows minimal variability in normalized fluorescence (see mean \pm SEM). To further address the Reviewer's point, we have added new Figure 2C, which shows the presence of the mNG::PLC δ 1-PH in the cilium proper of 100% of the *inpp-1* mutant animals. Thus, while it remains possible that there could be subtle differences in PIP₂ abundance in different RnB cilia, we are unable to conclude whether this underlies the cilium variable effect of the *inpp-1* mutant on ciliary PKD-2::GFP.

7) *Importantly, the distributions of PPK-1, INPP-1, and PIP2 in RnB cilia confirm that the PIP2 regulation described for other cilia types is conserved in these cilia, which are known for their high ectocytosis rate. This observation suggests that major differences in PIP2 regulation may not account for their high ectocytosis rate. Highlighting this point more clearly in the discussion would be valuable.*

Excellent point! We have now added a couple sentences to the Discussion (line 513) to highlight this observation.

8) Finally, the increased PIP2 in the cilium proper of *inpp-1* mutant and its effect on EV release does not explain the strong accumulation of PKD-2 in the dendrite of *inpp-1* and *inpp-5k*. It may be worth to present some potential explanations.

This is another excellent point; we have revised the Discussion (line 463) to indicate that disruption of the balance of phosphoinositide species could impact trafficking of PKD-2 to endocytic compartments, leading to the dendritic accumulation of PKD-2 observed in both the *inpp-1* and *inpp-5k* mutants.

Third decision letter

MS ID#: jcs.264005R2

MS Title: Phosphatidylinositol 4,5-bisphosphate Impacts Extracellular Vesicle Shedding from *C. elegans* Ciliated Sensory Neurons

Authors: Malek Elsawyid; Alexis Semmel; Krisha Parekh; Nahin Prova; Tao Ke; Jessica Tanis
Article Type: Research Article

Dear Dr Tanis,

I am happy to tell you that your manuscript has been accepted for publication in Journal of Cell Science, pending standard publication integrity checks.